# LANGUAGE REPRESENTATIONS CAN BE WHAT RECOMMENDERS NEED: FINDINGS AND POTENTIALS

**Leheng Sheng**[1]  **An Zhang**[1*]  **Yi Zhang**[2]  **Yuxin Chen**[1]  **Xiang Wang**[2]  **Tat-Seng Chua**[1]
[1]National University of Singapore   [2]University of Science and Technology of China
`leheng.sheng@u.nus.edu, anzhang@u.nus.edu, zy1230@mail.ustc.edu.cn,`
`e1143404@u.nus.edu, xiangwang1223@gmail.com, dcscts@nus.edu.sg`

## ABSTRACT

Recent studies empirically indicate that language models (LMs) encode rich world knowledge beyond mere semantics, attracting significant attention across various fields. However, in the recommendation domain, it remains uncertain whether LMs implicitly encode user preference information. Contrary to prevailing understanding that LMs and traditional recommenders learn two distinct representation spaces due to the huge gap in language and behavior modeling objectives, this work re-examines such understanding and explores extracting a recommendation space directly from the language representation space. Surprisingly, our findings demonstrate that item representations, when linearly mapped from advanced LM representations, yield superior recommendation performance. This outcome suggests the possible homomorphism between the advanced language representation space and an effective item representation space for recommendation, implying that collaborative signals may be implicitly encoded within LMs. Motivated by the finding of homomorphism, we explore the possibility of designing advanced collaborative filtering (CF) models purely based on language representations without ID-based embeddings. To be specific, we incorporate several crucial components (*i.e.,* a multilayer perceptron (MLP), graph convolution, and contrastive learning (CL) loss function) to build a simple yet effective model, with the language representations of item textual metadata (*i.e.,* title) as the input. Empirical results show that such a simple model can outperform leading ID-based CF models on multiple datasets, which sheds light on using language representations for better recommendation. Moreover, we systematically analyze this simple model and find several key features for using advanced language representations: a good initialization for item representations, superior zero-shot recommendation abilities in new datasets, and being aware of user intention. Our findings highlight the connection between language modeling and behavior modeling, which can inspire both natural language processing and recommender system communities.

## 1 INTRODUCTION

Language models (LMs) have achieved great success across various domains (Vaswani et al., 2017; Devlin et al., 2019; Dubey et al., 2024; OpenAI, 2023), raising a critical question about the knowledge encoded within the language space. Recent studies empirically find that LMs extend beyond semantic understanding to encode comprehensive world knowledge about various domains, such as game states (Li et al., 2023a), lexical attributes (Vulic et al., 2020), and even concepts of space and time (Gurnee & Tegmark, 2023) through language modeling. However, in the domain of recommendation where the integration of LMs is attracting widespread interest (Fan et al., 2023; Li et al., 2023b; Wu et al., 2023a), it remains unclear whether LMs inherently encode relevant information on user preferences and behaviors in the language space.

Currently, one prevailing understanding holds that general LMs and traditional recommenders (*e.g.,* collaborative filtering models (Koren et al., 2009; He et al., 2021)) encode distinct representation spaces — one for language space and the other for behavior space — but they offer the potential

---

*An Zhang is the corresponding author.

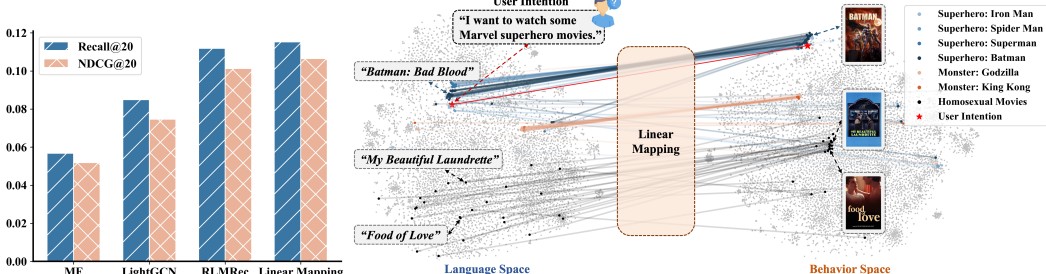

(a) Linearly mapping language representations into the behavior space for recommendation

(b) Performance comparison

(c) The t-SNE representations of movies and user intention in two spaces.

Figure 1: Linearly mapping item titles in language representation space into behavior space yields superior recommendation performance on Movies & TV (Ni et al., 2019) dataset. (1a) The framework of linear mapping. (1b) The recommendation performance comparison between leading CF recommenders and linear mapping. (1c) The t-SNE (Van der Maaten & Hinton, 2008) visualizations of movie representations, with colored lines linking identical movies or user intention across language space (left) and linearly projected behavior space for recommendation (right).

to enhance each other in downstream recommendation tasks (Liao et al., 2024). Specifically, on the one hand, when using LMs as recommenders to directly output items of interest, aligning the language space with the behavior space can significantly improve the recommendation performance (Lin et al., 2023a; Vats et al., 2024; Xu et al., 2024). Various alignment strategies are proposed, including fine-tuning LMs with user behavior data (Zhang et al., 2023e; Bao et al., 2023; Geng et al., 2022; Cui et al., 2022; Lin et al., 2023b), incorporating embeddings from traditional recommenders as a new modality of LMs (Liao et al., 2024; Zhang et al., 2023f; Yang et al., 2023; Tennenholtz et al., 2024), and extending the vocabulary of LMs with item tokens (Zhu et al., 2023; Zheng et al., 2023; Rajput et al., 2023; Zhai et al., 2024). On the other hand, when using LMs as the enhancer to represent item content (*e.g.,* text metadata), traditional recommenders greatly benefit from text embeddings (Yuan et al., 2022; 2023; Li et al., 2023c; Hou et al., 2024a; Liu et al., 2024b), semantic and reasoning information (Wei et al., 2024; Ren et al., 2024b; Xi et al., 2023), and generated user behaviors (Zhang et al., 2023b;d). Despite these efforts, explicit studies of the relationship between language and behavior spaces remain largely unexplored in the recommendation domain.

In this work, we re-examine this prevailing understanding, by exploring whether LM-generated language space has inherently encoded user preferences and behaviors. Specifically, we test the possibility of directly deriving a behavior space from the language space — that is, we assess whether the language representations of item text metadata (*e.g.,* titles) generated by LMs can independently predict user behaviors and achieve competitive recommendation performance. Positive results would imply that user behavioral patterns, such as collaborative signals (*i.e.,* users' preference on items reflected by their behavioral similarities) (Wang et al., 2019b), might be implicitly encoded by LMs. To test this hypothesis, we employ linear mapping (Merullo et al., 2023) to project language representations of item titles into a behavior space for recommendation, as Figure 1a shows. Our empirical observations and findings include:

- Before linear mapping, language representation similarities (*i.e.,* semantic textual similarities (STS) (Muennighoff et al., 2023)) may reflect user preference similarities for item contents. Considering Figure 1c as an example, movies with themes of superheroes and monsters cluster together in both language and behavior spaces.

- After linear mapping, language representations are transformed into high-quality behavior representations, which achieve exceptional recommendation performance, as Figure 1b and experimental results in Section 3.2 show. Moreover, the performance improves as the language model size increases and remains relatively robust to prompt disturbances (see Section 3.2).

- Post-mapping language representations encode user behaviroal similarities beyond STS. For instance, while certain movies, such as those of homosexual movies (illustrated in Figure 1c), show low STS and their representations disperse in the language space, their projections through linear mapping tend to cluster together, reflecting high user preference similarities.

These findings suggest the **homomorphism** (Dieudonne, 1969) between the LM-generated language space and an expressive behavior space for recommendation. Motivated by this insight, we explore the possibility of building advanced collaborative filtering (CF) models based solely on the language space. To be specific, considering text metadata solely as items' pre-existing features rather than the widely-used ID information, we perform a frozen LM to create the language representations of items; consequently, we apply a trainable projector (*i.e.,* a two-layer MLP with graph convolution) to map them into a behavior space, and then employ a contrastive loss (*i.e.,* InfoNCE (van den Oord et al., 2018; Wu et al., 2022)) to optimize. We term this model AlphaRec for its simplicity and a series of good properties. Surprisingly, our empirical results show that such a simple model can outperform leading ID-based CF models on multiple datasets. This result sheds light on using language representations for better recommendation.

Furthermore, we systematically analyze this simple model and discover several potentials of adopting language representations for recommendation. First, language representations may serve as a good initialization for item representations, with few adjustments to achieve high recommendation performance (see Section 5.1). This is evidenced by the rapid training convergence of AlphaRec. Second, advanced language representations provide strong zero-shot recommendation capability across entirely new datasets (see Section 5.2). By co-training on multiple datasets, AlphaRec can achieve performance comparable to or even surpassing the fully-trained LightGCN (He et al., 2021) on new datasets without additional training. This underscores the potential of adopting advanced language representations to develop more generalizable recommenders. Third, advanced language representations provide opportunities for perceiving user intentions to refine recommendation results (see Section 5.3). Endowed with the inherent semantic comprehension of language representations, AlphaRec can adjust recommendations according to text-based user intentions, enabling recommenders to evolve into intention-aware systems through a straightforward paradigm shift.

## 2 PRELIMINARY

### 2.1 TASK FORMULATION

Personalized recommendation aims to learn user preferences from historical behaviors (*i.e.,* historical interactions with items like view, click, purchase) and find items of interest to trigger users' future behaviors (Zhou et al., 2018). In this paper, we consider one common recommendation setting: collaborative filtering (CF) (Koren et al., 2022). It aims to select item $i \in \mathcal{I}$ that best matches user $u$'s preferences based on binary interaction behaviors $\mathbf{Y} = [y_{ui}]$, where $y_{ui} = 1$ indicates user $u \in \mathcal{U}$ has interacted with item $i$, and $y_{ui} = 0$ otherwise (Rendle, 2022). Scrutinizing leading CF models, we summarize a common paradigm $\hat{y}_{ui} = s \circ \phi_\theta(\mathbf{x}_u, \mathbf{x}_i)$ involving three components:

- For a user-item pair $(u, i)$, we first get their pre-existing features $\mathbf{x}_u$ and $\mathbf{x}_i$, which are usually set as ID information or one-hot encodings in CF (Koren et al., 2009; Rendle, 2022; He et al., 2021).
- Upon $\mathbf{x}_u$ and $\mathbf{x}_i$, the representation generation module $\phi$ parameterized by $\theta$ is adopted to transfer them into behavior representations $\mathbf{e}_u$ and $\mathbf{e}_i$, encoding the behavioral patterns of users. Its architecture can vary widely, including ID-based embeddings (Koren et al., 2009), multilayer perceptions (He et al., 2017), graph neural networks (Wang et al., 2019b; Cai et al., 2023), and variational autoencoders (Liang et al., 2018).
- Upon $\mathbf{e}_u$ and $\mathbf{e}_i$, the scoring function $s$ is used to quantify their relevance reflecting how likely user $u$ will interact with item $i$. One widely-used function is cosine similarity, $s(\mathbf{e}_u, \mathbf{e}_i) = \frac{\mathbf{e}_u^\top \mathbf{e}_i}{\|\mathbf{e}_u\| \cdot \|\mathbf{e}_i\|}$ (Chen et al., 2023; Wu et al., 2022).

### 2.2 ITEM REPRESENTATION GENERATION

Here we emphasize the critical role of item representation generation, which involves transforming item $i$'s pre-existing features $\mathbf{x}_i$ into representations $\mathbf{e}_i$ suitable for recommendation. This process is essential, as the quality of these representations directly impacts the recommendation performance.

In this paper, we focus mainly on two kinds of item representation generation tailor-made for different pre-existing features: ID- and LM-based generators.

**ID-based generator.** Prevailing CF models (Koren et al., 2009; Rendle, 2022; Koren et al., 2022; Wang et al., 2019b; He et al., 2021; Yu et al., 2024) typically convert the ID information of each item $i$ into one-hot encodings (*e.g.,* pre-existing features $\mathbf{x}_i$). These sparse features are then passed through a trainable generator, such as ID embedding matrices (Koren et al., 2009; Rendle, 2022) or optionally combined with graph convolution layers (Wang et al., 2019b; He et al., 2021; Yu et al., 2024), to generate dense representations $\mathbf{e}_i$. Optimizing the learning of ID-based representations allows the generator to effectively learn user preferences and behaviors, leading to competitive recommendation performance. However, such ID-based generators suffer several problems, such as poor domain transferability and lack of user intention-aware abilities, since one-hot encodings lack sufficient semantics beyond being identifiers (He et al., 2021).

**LM-based generator.** Beyond ID information, another research line (Pazzani & Billsus, 2007; Covington et al., 2016; Liu et al., 2024c; Zhang et al., 2024b; Liu et al., 2023b) explores using the text metadata of item $i$ (*e.g.,* titles, descriptions) as pre-existing features $\mathbf{x}_i$. These features are fed into the LM-based generator, typically a combination of two subsequent components: (1) A frozen LM to extract $i$'s language representation $\mathbf{z}_i$, such as the encoder-only LMs like BERT-style models (Devlin et al., 2019; Liu et al., 2019), the decoder-only LMs like Llama-style autoregressive models (Touvron et al., 2023b; Jiang et al., 2023) and OpenAI text embedding models (Neelakantan et al., 2022); and (2) A trainable projector to map $\mathbf{z}_i$ into the final representation $\mathbf{e}_i$, often using layers like graph convolution layers (He et al., 2021). Although such LM-based generators have been explored to enrich the item representations in literatures (Yuan et al., 2023; Li et al., 2023c; Ren et al., 2024b), few studies have demonstrated that they can solely outperform ID-based generators in recommendation tasks. Worse still, the relationship between the LM-based language space and the behavior space remain largely unexplored in the recommendation domain.

## 3 UNCOVERING COLLABORATIVE SIGNALS IN LMs VIA LINEAR MAPPING

In this section, we first explore the following research questions. **RQ1**: Do LMs inherently encode collaborative signals (*i.e.,* users' preferences for items as reflected by behavioral similarities) within their representation spaces? **RQ2**: If so, does the presence of such signals scale with model size, and are they robust across different settings? To investigate these questions, we use linear mapping to project language representations of item titles into a behavior space for recommendation. We detail the implementation of the linear mapping in Section 3.1. Subsequently, in Section 3.2, we empirically assess the existence and robustness of collaborative signals in language representations.

### 3.1 LINEAR MAPPING

Linear mapping is effective to study the representation properties of LMs (Merullo et al., 2023; Alain & Bengio, 2017), discovering the homomorphism (Dieudonne, 1969) between the language space and another space in the target domain. However, its application in the recommendation domain remains largely underexplored.

To bridge this gap, we train a linear mapping matrix $\boldsymbol{W}$ to project representations from the language space into a behavior space for recommendation. High performance of this linear mapping on the test set would indicate the presence of homomorphism between the language space and an effective behavior space, suggesting the possible existence of collaborative signals in the language representation space (Ravichander et al., 2021; Gurnee & Tegmark, 2023). The overall framework of linear mapping is illustrated in Figure 1a. Specifically, we use frozen LMs to transform pre-existing item title features $\mathbf{x}_i$ into language representations $\mathbf{z}_i$. To derive user representations, we compute the average of the language representations of the items a user $u$ has interacted with, denoted as $\mathbf{z}_u = \frac{1}{|\mathcal{N}_u|} \sum_{i \in \mathcal{N}_u} \mathbf{z}_i$, where $\mathcal{N}_u$ is the set of user $u$'s historical items. See Appendix B.2 for detailed procedures for obtaining language representations. The linear mapping matrix sets behavior representations of user $u$ and item $i$ as $\mathbf{e}_u = \boldsymbol{W}\mathbf{z}_u$ and $\mathbf{e}_i = \boldsymbol{W}\mathbf{z}_i$ respectively. To optimize the matrix $\boldsymbol{W}$, we adopt the InfoNCE loss (van den Oord et al., 2018) as the objective function, which has demonstrated strong performance in both ID-based (Zhang et al., 2023a; Yu et al., 2024) and LM-based generators (Ren et al., 2024b) (refer to equation 4 for the formula).

Table 1: The comparison of the recommendation performance of linear mapping with the classical ID-based CF baselines.

| | | Movies & TV | | | Video Games | | | Books | | |
|---|---|---|---|---|---|---|---|---|---|---|
| | | Recall | NDCG | HR | Recall | NDCG | HR | Recall | NDCG | HR |
| CF | MF (Rendle et al., 2012) | 0.0568 | 0.0519 | 0.3377 | 0.0323 | 0.0195 | 0.0864 | 0.0437 | 0.0391 | 0.2476 |
| | MultVAE (Liang et al., 2018) | 0.0853 | 0.0776 | 0.4434 | 0.0908 | 0.0531 | 0.2211 | 0.0722 | 0.0597 | 0.3418 |
| | LightGCN (He et al., 2021) | 0.0849 | 0.0747 | 0.4397 | 0.1007 | 0.0590 | 0.2281 | 0.0723 | 0.0608 | **0.3489** |
| Linear Mapping | BERT | 0.0415 | 0.0399 | 0.2362 | 0.0524 | 0.0309 | 0.1245 | 0.0226 | 0.0194 | 0.1240 |
| | RoBERTa | 0.0406 | 0.0387 | 0.2277 | 0.0578 | 0.0338 | 0.1339 | 0.0247 | 0.0209 | 0.1262 |
| | Llama2-7B | 0.1027 | 0.0955 | 0.4952 | 0.1249 | 0.0729 | 0.2746 | 0.0662 | 0.0559 | 0.3176 |
| | Mistral-7B | 0.1039 | 0.0963 | 0.4994 | 0.1270 | 0.0687 | 0.2428 | 0.0650 | 0.0544 | 0.3124 |
| | text-embedding-ada-v2 | 0.0926 | 0.0874 | 0.4563 | 0.1176 | 0.0683 | 0.2579 | 0.0515 | 0.0436 | 0.2570 |
| | text-embeddings-3-large | 0.1109 | 0.1023 | 0.5200 | 0.1367 | **0.0793** | **0.2928** | 0.0735 | 0.0608 | 0.3355 |
| | SFR-Embedding-Mistral | **0.1152** | **0.1065** | **0.5327** | **0.1370** | 0.0787 | 0.2927 | **0.0738** | **0.0610** | 0.3371 |

(a) Movie & TV            (b) Games            (c) Books

Figure 2: The recommendation performance of linear mapping with different language model sizes.

## 3.2 EMPIRICAL FINDINGS

**Existence (RQ1).** To explore the existence of collaborative signals in language representations, we test the recommendation performance of the linear mapping method. Table 1 reports the performance yielded by post-mapping representations on three Amazon datasets (Ni et al., 2019), comparing with classic ID-based CF baselines: matrix factorization (MF) (Koren et al., 2009), MultVAE (Liang et al., 2018), and LightGCN (He et al., 2021) (see baseline details in Appendix C.2). Figures (2a) - (2c) depict the linear mapping performance under different LM sizes. Figure 1c demonstrates the visualization of representations before and after linear mapping. We observe that:

- **Post-mapping representations of advanced LMs achieve superior recommendation performance in most cases, suggesting the possible homomorphism between language spaces and behavior spaces.** Specifically, advanced LMs (*e.g.,* Llama2-7B (Touvron et al., 2023b) and text-embeddings-3-large (Neelakantan et al., 2022)) consistently perform better than leading CF models (*e.g.,* LightGCN) on most metrics. We also see that the performance improves with more recent and advanced LMs. In contrast, earlier BERT-style models (*e.g.,* BERT (Devlin et al., 2019) and RoBERTa (Liu et al., 2019)) perform similarly to or worse than MF, indicating that LMs have only recently developed the ability to encode user preference similarities effectively.

- **Language representations encode user preference similarities beyond semantic textual similarities (STS).** Consider Figure 1c as an example again, homosexual movies, which differ significantly in textual meaning, cluster together after linear mapping. This suggests that user preferences, which are not immediately apparent from text alone, are implicitly encoded in the language space and can be uncovered through linear mapping.

**Scaling and Robustness (RQ2).** We also investigate whether the encoding of user preference similarities in LMs scales with model size and whether this encoding is robust in the presence of noise in the input prompts. To this end, we test the linear mapping performance of LMs of various sizes and prompts with noise:

- **The encoding of user preference similarities becomes more refined as model size increases, leading to better linear mapping performance.** Specifically, we test the linear mapping performance across different language model sizes (7B, 13B, and 70B) of the Llama2 family (Touvron et al., 2023b). Llama3 (Dubey et al., 2024) is not selected due to the lack of a 13B model. As shown in Figure 2, linear mapping performance improves consistently as the model size increases from 7B to 70B, indicating that larger models capture more nuanced user behavioral patterns.

Table 2: The robustness of language representations for recommendation.

| | Movies & TV | | | Video Games | | | Books | | |
|---|---|---|---|---|---|---|---|---|---|
| | Recall | NDCG | HR | Recall | NDCG | HR | Recall | NDCG | HR |
| Title + Random Noise | 0.0952 | 0.0887 | 0.4731 | 0.1213 | 0.0706 | 0.2722 | 0.0632 | 0.0525 | 0.3099 |
| Title Only | 0.1027 | 0.0955 | 0.4952 | 0.1249 | 0.0729 | 0.2746 | 0.0662 | 0.0559 | 0.3176 |

- **Language representations are relatively robust to prompt disturbances.** Following previous works (Gurnee & Tegmark, 2023), we compare two prompting strategies: using item titles alone (*e.g.,* Castlevania), and adding 5-10 random letters to the titles (*e.g.,* Castlevania sdfhdsk). Table 2 shows that adding random noise to the item titles had minimal impact on the linear mapping performance. The prompt noise has more impact on Movies & TV since item titles in this dataset are relatively shorter than others (see Appendix C.1). This finding suggests the relative robustness of the recommendation knowledge encoded in the language representation space.

# 4 LEVERAGING LANGUAGE REPRESENTATIONS FOR BETTER RECOMMENDATION

This finding of possible space homomorphism (Dieudonne, 1969) and encoded collaborative signals arouse interest in the following questions. **RQ3**: How powerful are such language representations for building advanced CF models that can outperform prevailing ID-based CF methods? To address these questions, in Section 4.1, we aim to develop a simple yet effective CF model termed AlphaRec, which is solely based on language representations and merely incorporates three crucial components in modern CF models. After that, we evaluate its performance in Section 4.2 to demonstrate the capability of advanced language representations for recommendation.

## 4.1 ALPHAREC

We briefly present how this simple model AlphaRec is designed and trained. It is important to highlight that we center on exploring the power of language representations for recommendation, rather than deliberately inventing new CF mechanisms. Generally, the representation generation architecture $\phi_\theta(\cdot, \cdot)$ is simple, which only contains a two-layer MLP and the basic graph convolution operation. The cosine similarity is used as the similarity function $s(\cdot, \cdot)$, and the contrastive loss InfoNCE (van den Oord et al., 2018; Wu et al., 2022) is adopted for optimization. For simplicity, we adopt text-embeddings-3-large (Neelakantan et al., 2022) for language representation generation by default, for its excellent language understanding and representation capabilities.

**Nonlinear projection.** We substitute the linear matrix delineated in Section 3 with a nonlinear MLP. Nonlinear transformation helps in excavating more comprehensive preference similarities from the language representation space (see discussions about this in Appendix C.7) (He et al., 2017). Taking the averaged language representaions of historical items as the user language representation (*i.e.,* $\mathbf{z}_u = \frac{1}{|\mathcal{N}_u|} \sum_{i \in \mathcal{N}_u} \mathbf{z}_i$), the initial nonlinear transformation operation be formulated as:

$$\mathbf{e}_i^{(0)} = \boldsymbol{W}_2 \operatorname{LeakyReLU}\left(\boldsymbol{W}_1 \mathbf{z}_i + \boldsymbol{b}_1\right) + \boldsymbol{b}_2, \quad \mathbf{e}_u^{(0)} = \boldsymbol{W}_2 \operatorname{LeakyReLU}\left(\boldsymbol{W}_1 \mathbf{z}_u + \boldsymbol{b}_1\right) + \boldsymbol{b}_2. \quad (1)$$

**Graph convolution.** Graph neural networks (GNNs) show superior effectiveness for recommendation (Wang et al., 2019b), owing to the natural user-item graph structure in recommender systems (Wu et al., 2023b). We employ a minimal graph convolution operation (He et al., 2021) to capture more complicated collaborative patterns from high-order connectivity (Wu et al., 2019) as follows:

$$\mathbf{e}_u^{(k+1)} = \sum_{i \in \mathcal{N}_u} \frac{1}{\sqrt{|\mathcal{N}_u|}\sqrt{|\mathcal{N}_i|}} \mathbf{e}_i^{(k)}, \quad \mathbf{e}_i^{(k+1)} = \sum_{u \in \mathcal{N}_i} \frac{1}{\sqrt{|\mathcal{N}_i|}\sqrt{|\mathcal{N}_u|}} \mathbf{e}_u^{(k)}. \quad (2)$$

The information of connected neighbors is aggregated with a symmetric normalization term $\frac{1}{\sqrt{|\mathcal{N}_u|}\sqrt{|\mathcal{N}_i|}}$. Here $\mathcal{N}_u$ ($\mathcal{N}_i$) denotes the historical item (user) set that user $u$ (item $i$) has interacted with. The representations $\mathbf{e}_u^{(0)}$ and $\mathbf{e}_i^{(0)}$ projected from the MLP are used as the input of the first layer. After propagating for $K$ layers, the final behavior representation of a user $u$ (item $i$) is obtained as the average of representations from each layer:

$$\mathbf{e}_u = \frac{1}{K+1} \sum_{k=0}^{K} \mathbf{e}_u^{(k)}, \quad \mathbf{e}_i = \frac{1}{K+1} \sum_{k=0}^{K} \mathbf{e}_i^{(k)}. \quad (3)$$

Table 3: The performance comparison with ID-based CF baselines. The improvement achieved by AlphaRec is significant ($p$-value $<< 0.05$).

| | Movies & TV | | | Video Games | | | Books | | |
|---|---|---|---|---|---|---|---|---|---|
| | Recall | NDCG | HR | Recall | NDCG | HR | Recall | NDCG | HR |
| MF (Rendle et al., 2012) | 0.0568 | 0.0519 | 0.3377 | 0.0323 | 0.0195 | 0.0864 | 0.0437 | 0.0391 | 0.2476 |
| MultVAE (Liang et al., 2018) | 0.0853 | 0.0776 | 0.4434 | 0.0908 | 0.0531 | 0.2211 | 0.0722 | 0.0597 | 0.3418 |
| LightGCN (He et al., 2021) | 0.0849 | 0.0747 | 0.4397 | 0.1007 | 0.0590 | 0.2281 | 0.0723 | 0.0608 | 0.3489 |
| SGL (Wu et al., 2021) | 0.0916 | 0.0838 | 0.4680 | 0.1089 | 0.0634 | 0.2449 | 0.0789 | 0.0657 | 0.3734 |
| BC Loss (Zhang et al., 2022) | 0.1039 | 0.0943 | 0.5037 | 0.1145 | 0.0668 | 0.2561 | 0.0915 | 0.0779 | 0.4045 |
| XSimGCL (Yu et al., 2024) | 0.1057 | 0.0984 | 0.5128 | 0.1138 | 0.0662 | 0.2550 | 0.0879 | 0.0745 | 0.3918 |
| XSimGCL$_t$ (Yu et al., 2024) | 0.1015 | 0.0951 | 0.5016 | 0.1199 | 0.0679 | 0.2674 | 0.0900 | 0.0736 | 0.4036 |
| KAR (Xi et al., 2023) | 0.1084 | 0.1001 | 0.5134 | 0.1181 | 0.0693 | 0.2571 | 0.0852 | 0.0734 | 0.3834 |
| RLMRec (Ren et al., 2024b) | 0.1119 | 0.1013 | 0.5301 | 0.1384 | 0.0809 | 0.2997 | 0.0928 | 0.0774 | 0.4092 |
| EMB-KNN | 0.0548 | 0.0380 | 0.2916 | 0.0879 | 0.0389 | 0.1970 | 0.0434 | 0.0248 | 0.1851 |
| **AlphaRec** | **0.1221\*** | **0.1144\*** | **0.5587\*** | **0.1519\*** | **0.0894\*** | **0.3207\*** | **0.0991\*** | **0.0828\*** | **0.4185\*** |
| Imp.% over the best baseline | 6.79% | 5.34% | 2.27% | 9.12% | 10.75% | 5.40% | 9.75% | 10.51% | 7.01% |

**Contrastive learning objective.** The introduction of contrastive learning (Radford et al., 2021) is another key element for the success of leading CF models. Recent research suggests that the contrast learning objective, rather than data augmentation, plays a more significant role in improving recommendation performance (Yu et al., 2024; 2022; Zhang et al., 2023a). Therefore, we simply use the contrast learning object InfoNCE (van den Oord et al., 2018) as the loss function without any additional data augmentation on the graph (Wu et al., 2022). With cosine similarity as the similarity function $s(\mathbf{e}_u, \mathbf{e}_i) = \frac{\mathbf{e}_u^\top \mathbf{e}_i}{\|\mathbf{e}_u\| \cdot \|\mathbf{e}_i\|}$, the InfoNCE loss (van den Oord et al., 2018) is written as:

$$\mathcal{L}_{\text{InfoNCE}} = - \sum_{(u,i) \in \mathcal{O}^+} \log \frac{\exp\left(s(\mathbf{e}_u, \mathbf{e}_i)/\tau\right)}{\exp\left(s(\mathbf{e}_u, \mathbf{e}_i)/\tau\right) + \sum_{j \in \mathcal{S}_u} \exp\left(s(\mathbf{e}_u, \mathbf{e}_j)/\tau\right)}. \tag{4}$$

Here, $\tau$ is a hyperparameter called temperature (Wang & Liu, 2021), $\mathcal{O}^+ = \{(u,i)|y_{ui} = 1\}$ denoting the observed interactions between users $\mathcal{U}$ and items $\mathcal{I}$. And $\mathcal{S}_u$ is a randomly sampled subset of negative items that user $u$ does not adopt.

## 4.2 EMPIRICAL FINDINGS

**Baselines.** We compare AlphaRec with leading ID-based CF baselines, to assess the effectiveness of adopting advanced language representations. In addition to classic baselines introduced in section 3.2, we consider two categories of leading ID-based CF baselines, CL-based CF methods: SGL (Wu et al., 2021), BC Loss (Zhang et al., 2022), XSimGCL (Yu et al., 2024) and LM-enhanced CF methods: KAR (Xi et al., 2023), RLMRec (Ren et al., 2024b). We also consider a variant of XSimGCL (*i.e.*, XSimGCL$_t$) where ID embeddings are replaced with language representations, and directly using language representations $\mathbf{z}_u$ and $\mathbf{z}_i$ for recommendation without training (*i.e.*, EMB-KNN). See baseline details in Appendix C.2.

**Recommendation capabilities (RQ3).** Table 3 presents performance comparison. To alleviate the possible information leakage problem that pretraining data of LMs may include the training data, we also conduct experiments on a newly published dataset Amazon 2023 in Appendix C.6. The best-performing methods are bold, while the second-best methods are underlined. We observe that:

- **Advanced language representations shows strong potentials for recommendation, which can be unleashed by appropriate model design**. AlphaRec consistently outperforms leading CF baselines by a large margin across all metrics on all datasets, with an improvement ranging from 6.79% to 9.75% on Recall@20 compared to the best baseline. Moreover, as shown in Figure 3a, each component contributes positively (see more ablation results and details in Appendix C.3) in unleashing the power of language representations. Moreover, simply replacing the ID embeddings in XSimGCL with language representations (*i.e.*, XSimGCL$_t$) does not bring a stable and remarkable improvement, which is consistent with previous findings (Yuan et al., 2023; Li et al., 2023c). This phenomenon highlights the importance of appropriate model design beyond simple input feature replacement. Above findings suggest that the power of advanced language representations can be unleashed by carefully designing the model, showcasing the potential to surpass prevailing ID-based recommenders.

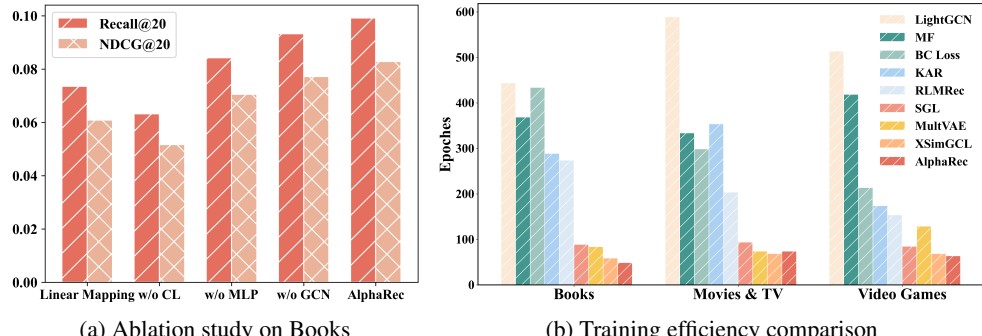

(a) Ablation study on Books        (b) Training efficiency comparison

Figure 3: (3a) The effect of each component on Books dataset. (3b) The number of epochs needed for each model to converge. AlphaRec exhibits a breakneck convergence speed.

- **The incorporation of advanced language representations can benefit traditional ID-based CF methods.** We note that two LM-enhanced CF methods, KAR and RLMRec, both show improvements over the most advanced CF methods. Nevertheless, the combination of ID-based embeddings and language representations in these methods does not yield higher results than purely language-representation-based AlphaRec. We attribute this phenomenon to their naive design for the combination ID-based embeddings and language representations, which is also highlighted by previous works (Yuan et al., 2023; Zhang et al., 2024d).

## 5 EXPLORING POTENTIALS OF LANGUAGE REPRESENTATIONS FOR RECOMMENDATION

In this section, we focus on this question: What new opportunities beyond good performance can advanced language representations bring to recommender systems? To answer this, we systematically analyze AlphaRec and discover the following potentials of adopting such language representations. **Potential 1:** Good initialization for item representations (Section 5.1). **Potential 2:** Zero-shot ability (Section 5.2). **Potential 3:** Intention-aware ability (Section 5.3).

### 5.1 GOOD INITIALIZATION FOR ITEM REPRESENTATIONS (POTENTIAL 1)

**Advanced language representations may provide a good initialization for item representations, with few adjustments for effective recommendation**. As shown in Figure 3b, beyond its good performance, AlphaRec also exhibits extremely fast convergence speed, which is comparable with or even surpasses the fastest ID-based CF methods (*e.g.,* SGL (Wu et al., 2021) and XSimGCL (Yu et al., 2024)). Moreover, recent works also suggest that, when using advanced language representations to initialize ID-based item embeddings (Harte et al., 2023; Zhao et al., 2024), the performance of traditional ID-based recommenders improves significantly. We attribute the above findings to the homomorphism between the language space and a good behavior space. Therefore, when using advanced language representations for initialization, only minor adjustments are needed to generate effective behavior representations for recommendation.

### 5.2 ZERO-SHOT ABILITY (POTENTIAL 2)

The prevailing ID-based recommenders suffer from domain transferring problems (*i.e.,* behavior representations are highly bound with ID information (Zhu et al., 2021)). Advanced language representations may provide opportunities for learning transferable item representations (Hou et al., 2022), enabling recommenders to perform well on entirely new datasets without any ID overlap. To address this potential, we test the zero-shot recommendation ability of AlphaRec (Ding et al., 2021).

**Experimental settings.** In zero-shot recommendation, there is no item or user overlap between the training set and test set (Ding et al., 2021; Zhang et al., 2024c), which is different from the research line of cross-domain recommendation (Zhu et al., 2021). We jointly train AlphaRec on three source datasets (*i.e.,* Books, Movies & TV, and Video Games), while testing it on three completely new target datasets (*i.e.,* Movielens-1M (Harper & Konstan, 2016), Book Crossing (Lee et al., 2019), and Amazon Industrial & Scientific (Ni et al., 2019)) without further training on these new datasets.

Table 4: The zero-shot recommendation performance comparison on entirely new datasets. The improvement achieved by AlphaRec is significant ($p$-value $<< 0.05$).

| | | Industrial & Scientific | | | MovieLens-1M | | | Book Crossing | | |
| | | Recall | NDCG | HR | Recall | NDCG | HR | Recall | NDCG | HR |
|---|---|---|---|---|---|---|---|---|---|---|
| full | MF (Rendle et al., 2012) | 0.0344 | 0.0225 | 0.0521 | 0.1855 | 0.3765 | 0.9634 | 0.0316 | 0.0317 | 0.2382 |
| | MultVAE (Liang et al., 2018) | 0.0751 | 0.0459 | 0.1125 | 0.2039 | 0.3741 | 0.9740 | 0.0736 | 0.0634 | 0.3716 |
| | LightGCN (He et al., 2021) | 0.0785 | 0.0533 | 0.1078 | 0.2019 | 0.4017 | 0.9715 | 0.0630 | 0.0588 | 0.3475 |
| zero-shot | Random | 0.0148 | 0.0061 | 0.0248 | 0.0068 | 0.0185 | 0.2611 | 0.0039 | 0.0036 | 0.0443 |
| | Pop | 0.0216 | 0.0087 | 0.0396 | 0.0253 | 0.0679 | 0.5439 | 0.0119 | 0.0101 | 0.1157 |
| | ZESRec (Ding et al., 2021) | 0.0326 | 0.0272 | 0.0628 | 0.0274 | 0.0787 | 0.5786 | 0.0155 | 0.0143 | 0.1347 |
| | UniSRec (Hou et al., 2022) | 0.0453 | 0.0350 | 0.0863 | 0.0578 | 0.1412 | 0.7135 | 0.0396 | 0.0332 | 0.2454 |
| | VQ-Rec (Hou et al., 2023) | 0.0645 | 0.0410 | 0.0963 | 0.0804 | 0.1921 | 0.8167 | 0.0485 | 0.0492 | 0.2825 |
| | **AlphaRec** | **0.0913*** | **0.0573** | **0.1277*** | **0.1486*** | **0.3215*** | **0.9296*** | **0.0660*** | **0.0545*** | **0.3381*** |
| | Imp.% over the best zero-shot baseline | 157.09% | 127.69% | 30.29% | 66.67% | 64.16% | 37.78% | 101.55% | 63.71% | 47.97% |

(see more details about training on multiple datasets in Appendix D.2.1). Due to the lack of zero-shot recommenders in general CF, we slightly modify three zero-shot methods in the sequential recommendation (Wang et al., 2019a), ZESRec (Hou et al., 2022), UniSRec (Hou et al., 2022), and VQ-Rec (Hou et al., 2023), as baselines. We also incorporate two strategy-based CF methods (*i.e.,* Random and Pop) and one method using the large language model (LLM) as zero-shot recommender (*i.e.,* LLMRank (Hou et al., 2024b)) (see more details about baselines in Appendix D.2.2).

**Findings.** Table 4 presents the zero-shot recommendation performance comparison. The best methods are bold and starred, while the second-best methods are underlined. We observe that:

- **Advanced language representations provide opportunities for learning transferable item representations.** AlphaRec demonstrates strong zero-shot recommendation capabilities, comparable to or even surpassing the fully trained LightGCN. AlphaRec performs better on the Amazon Industrial & Scientific dataset, possibly because it captures user behavioral patterns of the same platform (Ni et al., 2019) through training on multiple Amazon datasets. Conversely, ZESRec and UniSRec exhibit a marked performance decrement compared with AlphaRec. We attribute this phenomenon to two aspects. On the one hand, BERT-style LMs (Devlin et al., 2019; Liu et al., 2019) used in these works may not have effectively encoded user preference similarities, which is consistent with our previous findings in Section 3. On the other hand, components designed for the next item prediction task in sequential recommendation (Kang & McAuley, 2018) may not be suitable for capturing the general preferences of users in CF scenarios. Moreover, AlphaRec also outperforms the leading LLM-based zero-shot recommender LLMRank (see Appendix D.2.3).

- **The zero-shot recommendation capability of advanced language representations generally benefits from an increased amount of training data, without compromising performance on source datasets.** As illustrated in Table 12, the zero-shot performance of AlphaRec, when trained on a mixed dataset, is generally superior to training on one single dataset (Hou et al., 2022). Moreover, we discover that AlphaRec, when trained jointly on multiple datasets, hardly experiences a performance decline on each source dataset. These results indicate the general recommendation capability of a single pre-trained AlphaRec across multiple datasets. The above findings also offer a potential research path to achieve general recommendation capabilities, by incorporating more training data with more themes. See more details about these results in Appendix D.2.4.

## 5.3 INTENTION-AWARE ABILITY (POTENTIAL 3)

The language understanding ability in advanced language representations (especially representations from LLM-based text embedding models) offers the opportunity for perceiving text-based user intentions and refining recommendations. To study the potential of intention-aware ability, we introduce a new hyperparameter $\alpha$ in AlphaRec to combine user intentions with historical interests.

**Experimental settings.** To endow AlphaRec with user intention-aware ability, we adopt a simple paradigm shift by introducing a user intention representation $\mathbf{e}_u^{Intention}$. In the inference stage, we obtain the language representation $\mathbf{e}_u^{Intention}$ for each user intention query and combine it with the original user representation to get a new user representation as $\tilde{\mathbf{e}}_u^{(0)} = (1-\alpha)\mathbf{e}_u^{(0)} + \alpha\mathbf{e}_u^{Intention}$ (Ai et al., 2017). This new user representation $\tilde{\mathbf{e}}_u^{(0)}$ is sent into the pre-trained AlphaRec for recommendation. We test the user intention capture ability of AlphaRec on MovieLens-1M and Video Games. In the test set, only one target item remains for each user (Ai et al., 2017), with one intention query

Table 5: The performance comparison in user intention capture.

| | MovieLens-1M | | Video Games | |
|---|---|---|---|---|
| | HR@5 | NDCG@5 | HR@5 | NDCG@5 |
| TEM (Bi et al., 2020) | 0.2738 | 0.1973 | 0.2212 | 0.1425 |
| AlphaRec (w/o Intention) | 0.0793 | 0.0498 | 0.0663 | 0.0438 |
| AlphaRec (w Intention) | **0.4704\*** | **0.3738\*** | **0.2569\*** | **0.1862\*** |

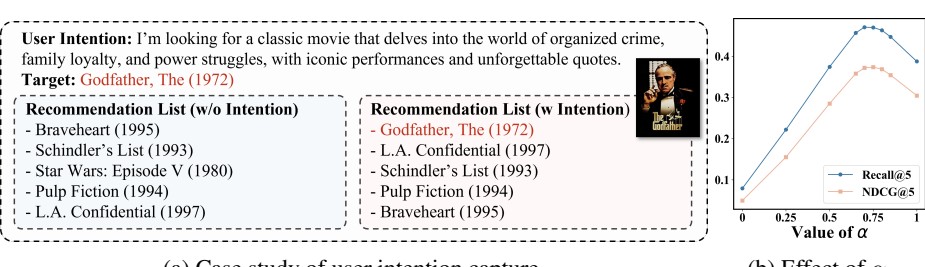

(a) Case study of user intention capture          (b) Effect of $\alpha$

Figure 4: User intention capture experiments on MovieLens-1M. (4a) AlphaRec refines the recommendations according to language-based user intention. (4b) The effect of user intention strength $\alpha$.

generated by ChatGPT (OpenAI, 2023; Hou et al., 2024a) (see the details about how to generate and check these intention queries in Appendix D.3.1). We report a relatively small $K = 5$ for all metrics to better reflect the intention capture accuracy.

**Findings:** We report the user intention capture experiment results in Table 5, show one case study in Figure 4a, and study the effect of $\alpha$ in Figure 4b. We find out that:

**The language understanding ability in advanced language representations enables recommenders to perceive user intentions and refine recommendations.** As shown in Table 5, the introduction of user intention (w Intention) significantly refines the recommendations of the pretrained AlphaRec (w/o Intention). Moreover, AlphaRec outperforms the baseline model TEM (Bi et al., 2020), even without additional training on search tasks. We further conduct a case study on MovieLens-1M to demonstrate how AlphaRec captures the user intention (see more examples in Appendix D.3.3). Additionally, the intention-aware ability benefits from user historical interests. Figure 4b depicts the effect of $\alpha$, which controls the strength of user intention. Here $\alpha = 0$ denotes that the user intention is neglected and $\alpha = 1$ denotes that the user historical interest is ignored. The convex curve in Figure 4b suggests that both user interests and user intention play vital roles. Above findings highlight the potential of adopting advanced language representations to perceive text-based user intentions and refine recommendations (see more details in Appendix D.3).

## 6 LIMITATIONS

There are several limitations unaddressed in this paper. First, this research lacks theoretical guarantee, and we do not contribute to designing any new components for CF models. Second, AlphaRec lacks a personalized design, since there is only a single MLP introduced for all users. The fixed user intention in Section 5.3 is less practical, lacking exploration on personalized user intentions.

## 7 CONCLUSION

In this paper, we explored the relationship between language space and behavior spaces for recommendation, and explored the potential for using language representations for recommendation. Empirical results suggest the possible presence of homomorphism between advanced LMs representation spaces and an effective item representation space for recommendation. Inspired by this finding, we discussed how to unleash the power of advanced language representations by developing a simple yet effective CF model called AlphaRec. Moreover, by systematically analyzing AlphaRec, we explored the potentials of advanced language representations: good initialization for item representations, zero-shot ability, and intention-aware ability. Possible future work will involve exploring the space relationship from both theoretical and multimodal perspectives. We believed that this paper sheds light on rethinking the connection between language modeling and user behavior modeling, benefiting both natural language processing and recommender system communities.

ACKNOWLEDGMENTS

This research is supported by the National Science and Technology Major Project (2023ZD0121102), the NExT Research Center, National Natural Science Foundation of China (92270114).

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

# A  RELATED WORKS

**Representations in LMs.** The impressive capabilities demonstrated by LMs across various tasks raise a wide concern about what they have learned in the representation space. Linear methods (*e.g.,* linear probing (Merullo et al., 2023) and linear mapping (Alain & Bengio, 2017)) are important and effective approaches for interpreting and analyzing representations of LMs (Ravichander et al., 2021). The main idea of linear methods is simple: training linear classifiers to predict some specific attributes or concepts (*e.g.,* lexical structure (Vulic et al., 2020) ) from the representations in the hidden layers of LMs, or transforming language representations into another feature space with a linear matrix. A high result of linear methods (*e.g.,* classification accuracy on the out-of-sample test set) tends to imply relevant information has been implicitly encoded in the representation space of LMs, although this does not imply LMs directly use these representations (Ravichander et al., 2021; Gurnee & Tegmark, 2023). Recent studies empirically demonstrate that concepts such as color (Patel & Pavlick, 2022), game states (Li et al., 2023a). and geographic position are encoded in LMs. Furthermore, these concepts may even be linearly encoded in the representation space of LMs (Li et al., 2023a; Park et al., 2023).

**Collaborative filtering.** Collaborative filtering (CF) (Ren et al., 2024a) is an advanced technique in modern recommender systems. The prevailing CF methods tend to adopt an ID-based paradigm, where users and items are typically represented as one-hot vectors, with an embedding table used for lookup (Koren et al., 2009). Usually, these embedding parameters are learned by optimizing specific loss functions to reconstruct the history interaction pattern (Rendle et al., 2012). Recent advances in CF mainly benefit from two aspects, graph convolution (Wu et al., 2023b) and contrastive learning (Ren et al., 2024a; Zhang et al., 2023c). These CF models exhibit superior recommendation performance by conducting the embedding propagation (Wang et al., 2019b; He et al., 2021) and applying contrastive learning objectives (Wu et al., 2021; Cai et al., 2023; Yu et al., 2024; Zhang et al., 2024a). However, although effective, these methods are still limited, due to the ID-based paradigm. Since one-hot vectors contain no feature information beyond being identifiers, it is challenging to transfer pre-trained ID embeddings to other domains (Hou et al., 2022) or to leverage leading techniques from computer vision (CV) and natural language processing (NLP) (Yuan et al., 2023).

**LMs for recommendation.** The remarkable language understanding and reasoning ability shown by LMs has attracted extensive attention in the field of recommendation. The application of LMs in recommendation can be categorized into three main approaches: LM-enhanced recommendation, LM as the modality encoder, and LM-as recommender. The first research direction, LM-enhanced recommendation, focuses on empowering traditional recommenders with the semantic representations from LMs (Xi et al., 2023; Ren et al., 2024b; Wei et al., 2024; Geng et al., 2024; Chen, 2023; Wang et al., 2024; Hou et al., 2023; Mao et al., 2023; Qiu et al., 2021; Zhang et al., 2024d; Liu et al., 2024a;d). Specifically, these methods introduce representations from LMs as additional features for traditional ID-based recommenders, to capture complicated user preferences. The second research line lies in adopting the LM as the text modality encoder, which is also known as a kind of modality-based recommendation (MoRec) (Yuan et al., 2023; Li et al., 2023c). These methods tend to train the LM as the text modality encoder together with the traditional recommender. In previous studies, BERT-style LMs are widely used as the text modality encoder. The third research line fails in directly using LMs as the recommender and recommends items in a text generation paradigm. Early attempts focus on adopting in-context learning (ICL) (Dong et al., 2022) and prompting pre-trained LMs (Hou et al., 2024b; Liu et al., 2023a; Dai et al., 2023; Gao et al., 2023). However, such naive methods tend to yield poor performance compared to traditional models. Therefore, recent studies concentrate on fine-tuning LMs on recommendation-related corpus (Bao et al., 2023; Zhang et al., 2023e; Lin et al., 2023b; Cui et al., 2022; Liu et al., 2024c; Hua et al., 2023; Chen et al., 2024; Ding et al., 2024) and align the LMs with the representations from traditional recommenders as the additional modality (Liao et al., 2024; Zhang et al., 2023f; Yang et al., 2023; Li et al., 2023d; Kong et al., 2024).

# B  UNCOVERING COLLABORATIVE SIGNALS IN LMS VIA LINEAR MAPPING

## B.1  BRIEF OF USED LMS

We briefly introduce the LMs we use for linear mapping in Section 3.1.

- **BERT** (Devlin et al., 2019) is an encoder-only language model based on the transformer architecture (Vaswani et al., 2017), pre-trained on text corpus with unsupervised tasks. BERT adopts bidirectional self-attention heads to learn bidirectional representations.

- **RoBERTa** (Liu et al., 2019) is an enhanced version of BERT. RoBERTa preserves the architecture of BERT but improves it by training with more data and large batches, adopting dynamic masking, and removing the next sentence prediction objective.

- **Llama2-7B** (Touvron et al., 2023b) is an open-source decoder-only LLM with 7 billion parameters. Llama2 adopts grouped-query attention, with longer context length and larger size of the pre-training corpus compared with Llama-7B (Touvron et al., 2023a).

- **Mistral-7B** (Jiang et al., 2023) is an open-source pre-trained decoder-only LLM with 7 billion parameters. Mistral 7B leverages grouped-query attention, coupled with sliding window attention for faster and lower cost inference.

- **text-embedding-ada-v2 & text-embeddings-3-large** (Neelakantan et al., 2022) are leading text embedding models released by OpenAI. These models are built upon decoder-only GPT models, pre-trained on unsupervised data at scale.

- **SFR-Embedding-Mistral** (Meng et al., 2024) is a decoder-based text embedding model built upon the open-source LLM Mixtral-7B (Jiang et al., 2023). SFR-Embedding-Mistral introduces task-homogeneous batching and computes contrastive loss on "hard negatives", which brings a better performance than the vanilla Mixtral-7B model.

### B.2 GENERATING ITEM REPRESENTATIONS FROM LMS

We present how to extract representations from LMs. For encoder-based LMs (*e.g.,* BERT (Devlin et al., 2019) and RoBERTa (Liu et al., 2019)), we use the representation of the last hidden state corresponding to the [CLS] token (Hou et al., 2024a). For decoder-based models (*e.g.,* Llama-7B (Touvron et al., 2023b; Jiang et al., 2023), Mistral-7B, and SFR-Embedding-Mistral (Meng et al., 2024)), we use the representation in the last transformer block (Vaswani et al., 2017), corresponding to the last input token (Gurnee & Tegmark, 2023; Todd et al., 2023; Neelakantan et al., 2022). Especially, for the commercial closed-source model (*e.g.,* text-embedding-ada-v2 and text-embeddings-3-large [1] (Neelakantan et al., 2022)), we directly call the API interface to obtain representations.

### B.3 CONNECTION WITH FINE-TUNED LLM-BASED APPROACHES FOR EXPLORING THE RELATIONSHIP BETWEEN LANGUAGE SPACE AND BEHAVIOR SPACE.

In exploring the space relationship between language modeling and user behavior modeling, there is another research line that uses LLMs to interpret behavior representations (*e.g.,* item representations trained from MF (Koren et al., 2009)) from traditional recommenders (Tennenholtz et al., 2024; Yang et al., 2023; Lei et al., 2024). They tend to project behavior representations into the token space of LLMs, and fine-tune LLMs to generate explanations for behavior representations (e.g., generate a movie description given a movie representation trained from MF). The fact that LLMs can interpret such behavior representations indicates that the behavior space and language space can be aligned at the token space level, which is similar to the findings in our paper. Our paper and these studies can be viewed as complementary works on a similar research goal (*i.e.,* exploration on the connection between language modeling and user behavior modeling from the representation space perspective), providing support for each other.

## C  LEVERAGING LANGUAGE REPRESENTATIONS FOR BETTER RECOMMENDATION

### C.1  DATASETS

We incorporate six datasets in this paper, including four datasets from the Amazon platform [2] (Ni et al., 2019) (*i.e.,* Books, Movies & TV, Video Games, and Industrial & Scientific), and two datasets from other platforms (*i.e.,* MovieLens-1M and Book Crossing). Table 6 reports the dataset statistics.

---

[1] https://platform.openai.com/docs/guides/embeddings
[2] www.amazon.com

Table 6: Dataset statistics.

| | Books | Movies & TV | Video Games | Industrial & Scientific | MovieLens-1M | Book Crossing |
|---|---|---|---|---|---|---|
| #Users | 71,306 | 26,073 | 40,834 | 15,141 | 6,040 | 6,273 |
| #Items | 26,073 | 12,464 | 14,344 | 5,163 | 3,043 | 5,335 |
| #Interactions | 2,209,030 | 876,027 | 390,013 | 82,578 | 995,492 | 253,057 |
| Density | 0.0008 | 0.0026 | 0.0007 | 0.0010 | 0.0542 | 0.0076 |

---

**Item Title Examples**

**Books:** *Dismissed with Prejudice: A J.P. Beaumont Novel*; *Die for Love: A Jacqueline Kirby Novel of Suspense*; *The Cloud*; *Memories Before and After the Sound of Music: An Autobiography*; *Harry Potter and the Sorcerer's Stone;*

**Movies & TV:** *Batman Begins*; *Fantastic Four*; *Max Headroom: The Complete Series*; *Madagascar*; *Land of the Dead*; *King Kong;*

**Video Games:** *USB Microphone for RockBand or Guitar Hero (PS3, Wii, Xbox360)*; *Command & Conquer: Tiberian Sun - PC*; *Tomb Raider III: Adventures of Lara Croft*; *Kartia: The Word of Fate*; *Snowboard Kids*; *Command & amp; Conquer: Tiberian Sun - PC*; *Final Fantasy VII*; *Grim Fandango - PC*; *Half-Life - PC;*

**MovieLens-1M:** *Basquiat (1996)*; *Tin Cup (1996)*; *Godfather, The (1972)*; *Supercop (1992)*; *Manny & Lo (1996)*; *Bound (1996)*; *Carpool (1996);*

**Book Crossing:** *Prague : A Novel*; *Chocolate Jesus*; *Wie Barney es sieht*; *To Kill a Mockingbird*; *Sturmzeit. Roman*; *A Soldier of the Great War*; *Pride and Prejudice (Dover Thrift Editions);*

**Industrial & Scientific:** *Jurassic Perisphinctes Ammonites from France*; *FS9140: Spinosaurus - Dinosaur Tooth 20-30mm*; *FS9410: USA Eocene, Fossil Fish (Knightia alt), A-grade*; *Delta 50-857 Charcoal Filter for 50-868*; *Hitachi RP30SA 7-1/2 Gallon Stainless Steel Industrial Shop Vacuum (Discontinued by Manufacturer)*; *Makita 632002-4 14-Inch Cut-Off Wheels (5-Pack) (Discontinued by Manufacturer)*; *PORTER-CABLE 740001801 4 1/2-Inch by 10yd 180 Grit Adhesive-Backed Sanding Roll*;

Figure 5: Example of item titles.

We divide the history interaction of each user into training, validation, and testing sets with a ratio of 4:3:3, and remove users with less than 20 interactions following previous studies (Zhang et al., 2023b). We also remove items from the test and validation sets that do not appear in the training set, to address the cold start problem.

In this paper, we only use the item titles as the text description. Figure 5 gives some item title examples from different datasets.

## C.2 BASELINES

We incorporate a series of ID-based CF models as our baselines for general recommendation. These models are classified as classical CF methods (MF, MultVAE, and LightGCN), CL-based CF methods (SGL, BC Loss, and XSimGCL), and LM-enhanced CF methods (KAR, RLMRec). We do not consider baselines using LMs as recommenders for two practical reasons: the huge inference cost on datasets with millions of interactions and the task limitation of candidate selection (Liao et al., 2024) or next item prediction (Zheng et al., 2023). For these LM-enhanced CF methods, we adopt the leading method XSimGCL as the backbone. We use the same item and user representations as we adopt in AlphaRec for KAR and XSimGCL, to make the comparison relatively fair. To evaluate the impact of the modifications, we demonstrate one ablation study on RLMRec in Appendix C.5.

- **MF** (Koren et al., 2009; Rendle et al., 2012) is the most basic CF model. It denotes users and items with ID-based embeddings and conducts matrix factorization with Bayesian personalized ranking (BPR) loss.

- **MultVAE** (Liang et al., 2018) is a traditional CF model based on the variational autoencoder (VAE). It regards the item recommendation as a generative process from a multinomial distribution

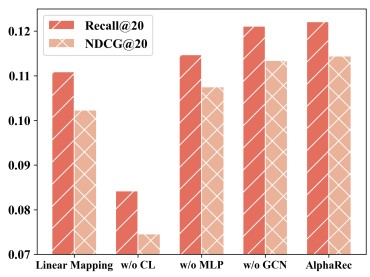
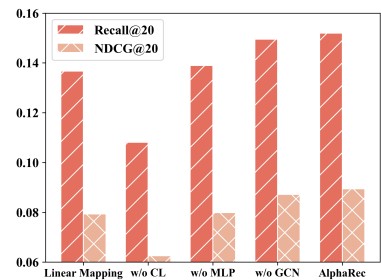

(a) Ablation study on Movies & TV      (b) Ablation study on Video Games

Figure 6: Ablation study

and uses variational inference to estimate parameters. We adopt the same model structure as suggested in the paper: $600 \rightarrow 200 \rightarrow 600$.

- **LightGCN** (He et al., 2021) is a light graph convolution network tailored for the recommendation, which deletes redundant feature transformation and activation function in NGCF (Wang et al., 2019b).

- **SGL** (Wu et al., 2021) introduces graph contrastive learning into recommender models for the first time. By employing node or edge dropout to generate augmented graph views and conduct contrastive learning between two views, SGL achieves better performance than LightGCN.

- **BC Loss** (Zhang et al., 2022) introduces a robust and model-agnostic contrastive loss, handling various data biases in recommendation, especially for popularity bias.

- **XSimGCL** (Yu et al., 2024) directly generates augmented views by adding noise into the inner layer of LightGCN without graph augmentation. The simplicity of XSimGCL leads to a faster convergence speed and better performance.

- **KAR** (Xi et al., 2023) enhances recommender models by integrating knowledge from LMs. It generates textual descriptions of users and items and combine the LM representations with traditional recommenders using a hybrid-expert adaptor.

- **RLMRec** (Ren et al., 2024b) aligns semantic representations of users and items with the representations in CF models through a contrastive loss, as an additional loss trained together with the CF model. The fusion of semantic information and collaborative information brings performance improvement.

## C.3 ABLATION STUDY

We conduct the same ablation study as introduced in Section 4 on Movies & TV and Video Games datasets. As illustrated in Figure 6, each component in AlphaRec contributes positively, which is consistent with our findings in Section 4. Specifically, the performance degradation caused by replacing the MLP with a linear weight matrix (w/o MLP) indicates that nonlinear transformations can extract the implicit user preference similarities encoded in the language representation space more effectively. Besides, the performance also drops from replacing InfoNCE loss (Wu et al., 2022) with BPR loss (Rendle et al., 2012) (w/o CL) and removing the graph convolution (w/o GCN) suggests that explicitly modeling the collaborative relationships through the loss function and model architecture can further enhance recommendation performance.

## C.4 EFFECT OF VARYING LANGUAGE REPRESENTATIONS IN ALPHAREC

We evaluate the effect of different language representations in AlphaRec by varying adopted LMs. As reported in Table 7, advanced language representations consistently yield higher performance than early BERT-style language representations. Moreover, due to the non-linearity and neighborhood aggregation introduced in AlphaRec, the performance gap between different language representations is narrowed.

Table 7: Effect of varying language representations in AlphaRec.

| | | Movies & TV | | | Video Games | | | Books | | |
|---|---|---|---|---|---|---|---|---|---|---|
| | | Recall | NDCG | HR | Recall | NDCG | HR | Recall | NDCG | HR |
| AlphaRec | BERT | 0.0994 | 0.0923 | 0.4873 | 0.0960 | 0.0550 | 0.2179 | 0.0719 | 0.0607 | 0.3434 |
| | RoBERTa | 0.0967 | 0.0895 | 0.4793 | 0.0947 | 0.0545 | 0.2167 | 0.0710 | 0.0596 | 0.3386 |
| | Llama2-7B | 0.1160 | 0.1092 | 0.5388 | 0.1395 | 0.0817 | 0.3003 | 0.0940 | 0.0793 | 0.4081 |
| | Mistral-7B | 0.1161 | 0.1097 | 0.5421 | 0.1413 | 0.0828 | 0.3020 | 0.0945 | 0.0799 | 0.4090 |
| | text-embedding-ada-v2 | 0.1152 | 0.1083 | 0.5382 | 0.1437 | 0.0844 | 0.3062 | 0.0933 | 0.0784 | 0.4061 |
| | text-embeddings-3-large | 0.1221 | **0.1144** | **0.5587** | 0.1519 | **0.0894** | 0.3207 | **0.0991** | **0.0828** | **0.4185** |
| | SFR-Embedding-Mistral | **0.1225** | 0.1139 | 0.5571 | **0.1521** | 0.0887 | **0.3209** | 0.0982 | 0.0820 | 0.4161 |

Table 8: Performance comparison of different versions of RLMRec .

| | Movies & TV | | | Video Games | | | Books | | |
|---|---|---|---|---|---|---|---|---|---|
| | Recall | NDCG | HR | Recall | NDCG | HR | Recall | NDCG | HR |
| XSimGCL | 0.1057 | 0.0984 | 0.5128 | 0.1138 | 0.0662 | 0.2550 | 0.0879 | 0.0745 | 0.3918 |
| RLMRec (LLM-generated profile) | 0.1046 | 0.0942 | 0.5063 | 0.1218 | 0.0696 | 0.2692 | 0.0905 | 0.0741 | 0.4049 |
| RLMRec (This paper) | 0.1119 | 0.1013 | 0.5301 | 0.1384 | 0.0809 | 0.2997 | 0.0928 | 0.0774 | 0.4092 |

## C.5 ABLATION STUDY ON THE MODIFICATION OF RLMREC

We have modified RLMRec (Ren et al., 2024b) by using the same item and user representations as in AlphaRec to make the comparison more fair, rather than using representations of LLM-generated item and user profiles. To evaluate the effectiveness of this modification, we conduct an ablation study on the adopted item and user representations. Specifically, we implement another version of RLMRec (Ren et al., 2024b) by using the original prompting approach to generate item and user profile, and name this version as RLMRec (LLM-generated profile). As shown in Table 8, the performance of RLMRec (LLM-generated profile) is lower than the implementation in this paper. Furthermore, the performance of incorporating LLM-generated profiles is not always stable, which may even lead to performance degradation in some cases (e.g., Movies & TV). We attribute the performance gap to the possible noise and hallucination introduced by these generated profiles.

## C.6 PERFORMANCE ON AMAZON ELECTRONIC 2023

To alleviate the information leakage problem that the early datasets may be used in the pretraining data of LMs, we consider the latest version of Amazon 2023 (Hou et al., 2024a) as the newly published dataset. This dataset was published in March 2024, which is the latest public dataset that we have access to. To further alleviate the information leakage problem, we only consider the interactions after 2022. We conduct the same experiment as in Section 4 on this new dataset, and present the performance in Table 9. As shown in this table, in this new dataset, AlphaRec still consistently outperforms baselines. We attribute this to the fact that language models understand the user preference behind item title description, rather than naively remembering the training data. Therefore, AlphaRec still works fine on the latest dataset.

## C.7 THE T-SNE VISUALIZATION COMPARISON

In this section, we aim to intuitively explore how the MLP in AlphaRec further helps in excavating collaborative signals in language representations, compared to the linear mapping matrix. We visualize the item representations from LMs, post-mapping representations from AlphaRec (w/o MLP), and post-mapping representations from AlphaRec in Figure 7, where AlphaRec (w/o MLP) denotes replacing the MLP with a linear mapping matrix. We observed that movies about superhero and monster cluster in all representation spaces, indicating both AlphaRec (w/o MLP) and AlphaRec capture the preference similarities between these items and preserve the clustering relationship. The difference between AlphaRec (w/o MLP) and AlphaRec lies in the ability to capture obscure preference similarities among items. As shown in Figure 7a, homosexual movies are dispersed in the language space, indicating the possible semantic differences between them. AlphaRec successfully captures the preference similarities and gathers these items in the representation space, while AlphaRec (w/o MLP) remains some items dispersed. Moreover, AlphaRec outperforms AlphaRec (w/o MLP) by a large margin, as indicated in Figure 6a. These results indicate that AlphaRec exhibits a more fine-grained preference capture ability with the help of nonlinear transformation.

Table 9: Performance of AlphaRec on Amazon Electronics 2023

|  | Amazon Electronics 2023 | | |
| --- | --- | --- | --- |
|  | Recall | NDCG | HR |
| MF (Rendle et al., 2012) | 0.0130 | 0.0089 | 0.0136 |
| MultVAE (Liang et al., 2018) | 0.0227 | 0.0158 | 0.0237 |
| LightGCN (He et al., 2021) | 0.0237 | 0.0161 | 0.0248 |
| SGL (Wu et al., 2021) | 0.0519 | 0.0250 | 0.0551 |
| BC Loss (Zhang et al., 2022) | 0.0548 | 0.0265 | 0.0585 |
| XSimGCL (Yu et al., 2024) | 0.0534 | 0.0261 | 0.0569 |
| KAR (Xi et al., 2023) | 0.0611 | 0.0283 | 0.0661 |
| RLMRec (Ren et al., 2024b) | 0.0633 | 0.0288 | 0.0674 |
| **AlphaRec** | **0.0687\*** | **0.0323\*** | **0.0732\*** |

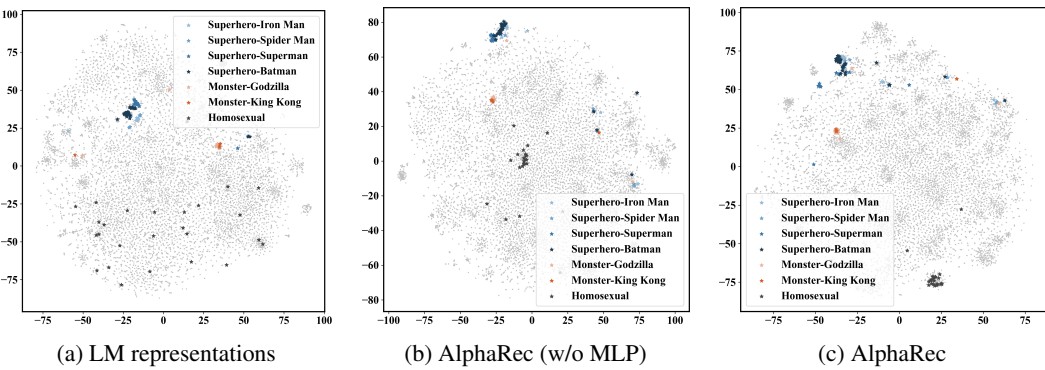

(a) LM representations        (b) AlphaRec (w/o MLP)        (c) AlphaRec

Figure 7: The t-SNE visualization of representations on Movies & TV. (7a) The item representations in the LM space. (7b) The item representations obtained by replacing the MLP with a linear mapping matrix in AlphaRec. (7c) The item representations obtained from AlphaRec.

# D EXPLORING POTENTIALS OF LANGUAGE REPRESENTATIONS FOR RECOMMENDATION

## D.1 FAST CONVERGENCE SPEED

We report the training cost of AlphaRec in this section. Table 10 reports the seconds needed per epoch and the total training cost until convergence. Here Amazon-Mix denotes the mixed dataset of Books, Movies & TV, and Video Games. It's worth noting that AlphaRec converges quickly and only requires a small amount of training time.

## D.2 ZERO-SHOT ABILITY

### D.2.1 CO-TRAINING ON MULTIPLE DATASETS

Co-training on multiple datasets is similar to training on one single dataset, where the only difference lies in the negative sampling. When co-training on multiple datasets, the negative items are restricted to the same dataset as the positive item rather than the full item pool. The other training procedures remain the same with training on one single dataset.

### D.2.2 BASELINES

Since previous works about zero-shot recommendation mostly focus on sequential recommendation (Kang & McAuley, 2018; Wang et al., 2019a), we slightly modify three methods in sequential recommendation, ZESRec (Ding et al., 2021), UniSRec (Hou et al., 2022), and VQ-Rec (Hou et al., 2023) as our baselines. Specifically, we maintain the model structure as provided in the paper, and adopt the training paradigm of CF.

- **Random** denotes randomly recommending items from the entire item pool.

Table 10: Training cost of AlphaRec (seconds per epoch/in total).

| | Books | Movies & TV | Video Games | Amazon-Mix |
|---|---|---|---|---|
| AlphaRec | 40.1 / 1363.4 | 12.3 / 479.7 | 7.4 / 214.6 | 107.2 / 5788.8 |

- **Pop** denotes randomly recommending from the most popular items. Here popularity denotes the number of users that have interacted with the item.

- **ZESRec** (Ding et al., 2021) is the first work that defines the problem of zero-shot recommendation. To address this problem, this work introduces a hierarchical Bayesian model with representations from the pre-trained BERT.

- **UniSRec** (Hou et al., 2022) aims to learn universal item representations from BERT, with parametric whitening and a MoE-enhanced adaptor. By pre-training on multiple source datasets, UniSRec can conduct zero-shot recommendation on various datasets in a transductive or inductive paradigm.

- **VQ-Rec** (Hou et al., 2023) learns vector-quantized representations for items, enabling transferring between datasets. A codebook is learned from language representations, for looking up item embeddings.

### D.2.3 COMPARISON WITH LLMRANK

Table 11: Zero-shot performance comparison with LLMRank

| | MovieLens-1M | | | | Steam | | | |
|---|---|---|---|---|---|---|---|---|
| | NDCG@1 | NDCG@5 | NDCG@10 | NDCG@20 | NDCG@1 | NDCG@5 | NDCG@10 | NDCG@20 |
| LLMRank | 0.2485 | 0.4115 | 0.5249 | 0.5612 | 0.3112 | 0.4413 | 0.5255 | 0.5302 |
| AlphaRec | 0.3919 | 0.6038 | 0.6543 | 0.6672 | 0.4450 | 0.6131 | 0.6394 | 0.6714 |
| Imp .% | 57.71% | 46.73% | 24.65% | 18.89% | 42.99% | 38.93% | 21.67% | 26.63% |

Table 11 illustrates the zero-shot recommendation performance compared with the LLM4Rec method LLMRank. We adopt the same setting of LLMRank, equipping 19 negative items for each positive item, and evaluate the NDCG on the candidate set. AlphaRec exhibits excellent zero-shot performance, significantly surpassing LLMRank. Moreover, the improvement over LLMRank exhibits a rising trend as the $K$ of NDCG decreases.

### D.2.4 THE EFFECT OF TRAINING DATASETS

**The effect of the training dataset scale on zero-shot recommendation.** We report the zero-shot recommendation performance differences trained on different datasets in Table 12. Here AlphaRec (trained on Books) denotes training on a single Books dataset, while AlphaRec (trained on mixed dataset) denotes co-training on three Amazon datasets. Generally, training on more datasets leads to a better zero-shot performance. In addition, we observe that, for the zero-shot performance on untrained target datasets, training datasets with similar themes contribute more (*e.g.,* Movies & TV and MovieLens-1M).

**The performance comparison between training on the single dataset and the mixed dataset.** In Table 13, AlphaRec (trained on single dataset) denotes training and testing on the same single dataset, while AlphaRec (trained on mixed dataset) denotes training on three Amazon datasets (*i.e.,* Books, Movies & TV, and Video Games) and testing on one single dataset. Generally, co-training on three Amazon datasets yields similar performance compared with training on one single dataset. The only exception lies in Video Games, which shows some performance degradation. We attribute this to the difference between the selection of $\tau$. We use $\tau = 0.15$ when trained on the mixed dataset, while the optimal $\tau$ for Video Games lies around 0.2. These results indicate that a single AlphaRec can capture user preferences among various datasets, showcasing a general collaborative signal capture ability.

Table 12: The effect of the training dataset on zero-shot recommendation

|  | Industrial & Scientific | | | MovieLens-1M | | | Book Crossing | | |
|---|---|---|---|---|---|---|---|---|---|
|  | Recall | NDCG | HR | Recall | NDCG | HR | Recall | NDCG | HR |
| **AlphaRec (trained on Books)** | 0.0896 | 0.0562 | 0.1256 | 0.1218 | 0.2619 | 0.8942 | 0.0646 | 0.0532 | 0.3346 |
| **AlphaRec (trained on Movies & TV)** | 0.0909 | **0.0581** | 0.1266 | 0.1438 | 0.3122 | 0.9200 | 0.0471 | 0.0406 | 0.2600 |
| **AlphaRec (trained on Video Games)** | 0.0905 | 0.0567 | 0.1225 | 0.1221 | 0.2313 | 0.9034 | 0.0412 | 0.0378 | 0.2585 |
| **AlphaRec (trained on mixed dataset)** | **0.0913** | 0.0573 | **0.1277** | **0.1486** | **0.3215** | **0.9296** | **0.0660** | **0.0545** | **0.3381** |

Table 13: Performance comparison between training on the single dataset and the mixed dataset

|  | Books | | | Movies & TV | | | Video Games | | |
|---|---|---|---|---|---|---|---|---|---|
|  | Recall | NDCG | HR | Recall | NDCG | HR | Recall | NDCG | HR |
| **AlphaRec (trained on single dataset)** | **0.0991** | **0.0828** | **0.4185** | **0.1221** | **0.1144** | **0.5587** | **0.1519** | **0.0894** | **0.3207** |
| **AlphaRec (trained on mixed dataset)** | 0.0979 | 0.0818 | 0.4147 | 0.1194 | 0.1107 | 0.5463 | 0.1381 | 0.0827 | 0.2985 |

## D.3 Intention-aware Ability

### D.3.1 Intention Query Generation

The user intention query is a natural language sentence implying the target item of interest. For each item in the dataset, we generate a fixed user intention query. Following the previous work (Hou et al., 2024a), we generate user intention queries with the help of ChatGPT (OpenAI, 2023). As shown in Figure 8, we prompt ChatGPT in a Chain-of-Thought (CoT) (Wei et al., 2022) paradigm and adopt the output as the user intention query. We adopt a rule-based strategy to ensure the quality of generated queries, and regenerate the wrong query. Considering the huge amount of item title text, we use ChatGPT3.5 API for generating all queries for the budget's sake.

### D.3.2 Baseline

AlphaRec exhibits user intention capture abilities, although not specially designed for search tasks. We compare AlphaRec with TEM (Bi et al., 2020) which falls in the field of personalized search (Ai et al., 2017; McAuley et al., 2015).

• **TEM** (Bi et al., 2020) uses a transformer to encode the intention query together with user history behaviors, which enables it to achieve better search results by considering the user's historical interest.

### D.3.3 Case Study

We conduct two more case studies to verify the user intention capture ability of AlphaRec. As illustrated in Figure 9 and Figure 10, AlphaRec provides better recommendation results, assigning the target item at the top while maintaining the general user preferences.

### D.3.4 Effect of the Intention Strength Alpha

The value of $\alpha$ controls the balance between the user's historical interests and the user intention query. A larger $\alpha$ incorporates more about the user intention while considering less about the user's historical interests. As shown in Figure 11, the effect of $\alpha$ on Video Games shows a similar trend with MovieLens-1M.

## E  Hyperparameter Settings and Implementation Details

We conduct all the experiments in PyTorch with a single NVIDIA RTX A5000 (24G) GPU and a 64 AMD EPYC 7543 32-Core Processor CPU. We optimize all methods with the Adam optimizer. For all ID-based CF methods, we set the layer numbers of graph propagation by default at 2, with the embedding size as 64 and the size of sampled negative items $|\mathcal{S}_u|$ as 256. We use the early stop strategy to avoid overfitting. We stop the training process if the Recall@20 metric on the validation set does not increase for 20 successive evaluations. In AlphaRec, the dimensions of the input and

---

**Intention Query Generation**

**Input**
You are an expert in generating queries for a target movie. Please help me generate the most suitable query for the target movie within one sentence, following the given example.
Example:
TARGET: BUG-A-SALT 3.0 Black Fly Edition.
QUERY: I want a gun that I can use while gardening to get rid of stink bugs, ants, flies, and spiders in my house. It needs to be amazing and help me feel less scared.
TARGET: Toy Story (1995).

**Output**
QUERY: I'm looking for a heartwarming animated movie that follows the adventures of a group of toys who come to life when their owner is not around.

---

Figure 8: Example of item query generation.

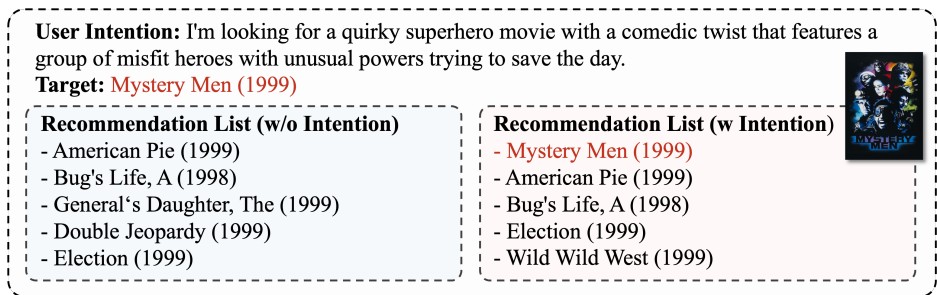

Figure 9: Case study of user intention capture on MovieLens-1M

output in the two-layer MLP are 3072 and 64 respectively, with the hidden layer dimension as 1536. We apply the all-ranking strategy (Krichene & Rendle, 2020) for all experiments, which ranks all items except positive ones in the training set for each user. We search hyperparameters for baselines according to the suggestion in the literature. The hyperparameter search space is reported in Table 14. For these LM-enhanced models, KAR and RLMRec, we also search the hyperparameter of their backbone XSimGCL.

For AlphaRec, the only hyperparameter is the temperature $\tau$ and we search it in [0.05, 2]. We report the temperature $\tau$ we used for each dataset in Table 15. For the mixed dataset Amazon-Mix in Section 5.2, we use a universal $\tau = 0.15$. We adopt $\tau = 0.2$ for the MovieLens-1M dataset for the user intention capture experiment in Section 5.3.

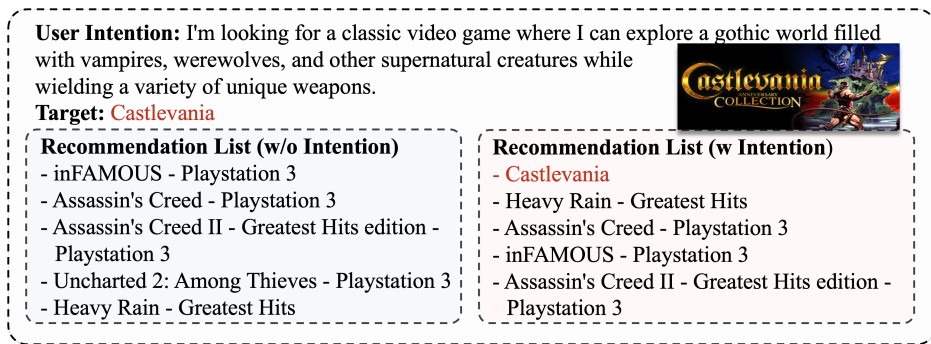

Figure 10: Case study of user intention capture on Video Games

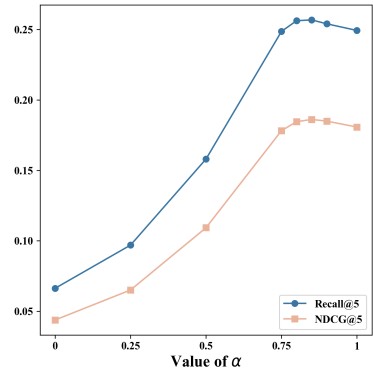

Figure 11: Effect of $\alpha$ on Video Games

Table 14: Hyperparameters search space for baselines.

| | Hyperparameter space |
|---|---|
| **MF** & **LightGCN** | lr $\sim$ {1e-5, 3e-5, 5e-5, 1e-4, 3e-4, 5e-4, 1e-3} |
| **MultVAE** | dropout ratio $\sim$ {0, 0.2, 0.5}, $\beta \sim$ {0.2, 0.4, 0.6, 0.8} |
| **SGL** | $\tau \sim$ [0.05, 2], $\lambda_1 \sim$ {0.005, 0.01, 0.05, 0.1, 0.5, 1.0}, $\rho \sim$ {0, 0.1, 0.2, 0.3, 0.4, 0.5} |
| **BC Loss** | $\tau_1 \sim$ [0.05, 3], $\tau_2 \sim$ [0.05, 2] |
| **XSimGCL** | $\tau \sim$ [0.05, 2], $\epsilon \sim$ {0.01, 0.05, 0.1, 0.2, 0.5, 1.0}, $\lambda \sim$ {0.005, 0.01, 0.05, 0.1, 0.5, 1.0}, $l* = 1$ |
| **KAR** | No. shared experts $\sim$ {3, 4, 5}, No. preference experts $\sim$ {4, 5} |
| **RLMRec** | kd weight $\sim$ [0.05, 2], kd temperature $\sim$ [0.01, 0.05, 0.1, 0.15, 0.2, 0.5, 1] |
| **ZESRec** | $\lambda_u \sim$ {0.01, 0.05, 0.1, 0.5, 1.0}, $\lambda_v \sim$ {0.01, 0.05, 0.1, 0.5, 1.0} |
| **UniSRec** | lr $\sim$ {3e-4, 1e-3, 3e-3, 1e-2} |
| **TEM** | $l \sim$ {2,3}, head $h \sim$ {4, 8} |
| **AlphaRec** | $\tau \sim$ [0.05, 2] |

Table 15: The hyperparameters of AlphaRec

| | Books | Movies & TV | Video Games | Amazon-Mix |
|---|---|---|---|---|
| $\tau$ | 0.15 | 0.15 | 0.2 | 0.15 |

Table 16: Cost for extracting language representations on Amazon Movie & TV

| per 10,000 items | Time Cost | Money Cost |
|---|---|---|
| Llama2-7B | 5 mins | 0 $ |
| text-embedding-3-large | 40 s | 0.17 $ |

