# OpenReview forum: "Language Representations Can be What Recommenders Need: Findings and Potentials"
_ICLR.cc/2025/Conference — ICLR 2025 Oral_

### Official Review · Reviewer_m7fA · 2024-10-16

**Soundness:** 2
**Presentation:** 4
**Contribution:** 3
**Rating:** 8
**Confidence:** 5

**Summary:**

This paper explores the relationship between the language space generated by LLMs and the behavior space produced by CF models. The authors found that the two spaces exhibit a pattern of homomorphism. Based on this finding, they proposed AlphaRec, which uses item embeddings exclusively from the language space and employs a trainable projector to map them into the behavior space for recommendations. The results demonstrate that with language space-initialized embeddings, the model significantly outperforms traditional ID-based methods and other LLM-based approaches in general recommendation, zero-shot recommendation, and intention-aware recommendation.

**Strengths:**

- The questions explored in this paper are novel and meaningful. While many existing works have analyzed the recommendation capabilities of language models, very few have empirically investigated the differences between the representation spaces of LMs and CF models.

- The evaluations are comprehensive. The authors not only investigate the recommendation capabilities of AlphaRec but also examine its performance in zero-shot and intention-aware recommendations. Additionally, the visualization of the LM and behavior spaces is a plus.

**Weaknesses:**

- There are some concerns that affect the soundness and the conclusions presented in the paper (see point 2 of weakness and questions below).

- Some of the comparisons are not entirely fair. For example, in Figure 3 and Table 3, it is evident that when using BPR loss (which is used in LightGCN), AlphaRec’s performance is a bit worse than that of LightGCN. Given that AlphaRec shares a similar graph convolution architecture with LightGCN in its nonlinear projection, it appears that AlphaRec’s advantage over LightGCN comes from the InfoNCE loss rather than the language representations. I am wondering whether such result can support the power of language representations. By the way, I suggest aligning the implementations of KAR and RLMRec closer with AlphaRec, including the loss function and some of the modules. Without a fair comparison, it is hard to conclude that the combination of ID-based embeddings and language representations is less suitable than using pure language representations.

**Questions:**

- Since MovieLens and Amazon Review are classical and widely used datasets publicly available on the Internet, I am concerned that they might already be part of the pretraining data for some LLMs, hence the user preference information encoded in LMs could be a potential data leakage. This could explain why BERT-style models perform worse than more recent LMs. To verify this assumption, can authors evaluate AlphaRec’s performance on a private or newly published recommendation dataset?

- If authors want to claim the collaborative signals are already encoded in the language space, I wonder if authors can show the results using original language representations without any linear or nonlinear transformations? If there is valid information can show user behaviour in language space, the performance should be comparable to some traditional CF methods (like MF) or heuristic methods (e.g., Pop).

---

> ### Author Response · Authors · 2024-11-23
> **Response to reviewer m7FA - Part (1/3)**
>
> We would like to thank you for your acknowledgment of our research topic and your valuable suggestions.
>
> ---
>
> > **Comment 1: The performance gain seems come from the InfoNCE loss rather than language representations. Whether such a result can support the power of language representations.**
>
> Thanks for your careful reading and for raising this important question.
> We agree with your observation and would like to clarify a key point: **naively incorporating language representations does not guarantee better in-domain performance in recommendations when compared with advanced ID-based recommenders, highlighting the importance of appropriate model design.**
>
> AlphaRec’s seemingly simple design plays a critical role in achieving its performance, highlighting its unique contribution compared to the linear mapping version.
>
> To better address your concern, we conducted additional experiments on the Movies & TV, Video Games, and Books datasets with six baselines: (ID-based) LightGCN+BPR, LightGCN+InfoNCE; (contrastive learning-based) XSimGCL, XSimGCL$ _{t}$  (replacing the ID embeddings in XSimGCL with language representations); AlphaRec+BPR, AlphaRec+InfoNCE.
> The results are illustrated in the following table.
>
>
> |                    |            | Movies & TV |            |            | Video Games |            |            | **Books**  |            |
> | :----------------: | :--------: | :---------: | :--------: | :--------: | :---------: | :--------: | :--------: | :--------: | :--------: |
> |                    |   Recall   |    NDCG     |     HR     |   Recall   |    NDCG     |     HR     |   Recall   |    NDCG    |     HR     |
> |   LightGCN + BPR   |   0.0849   |   0.0747    |   0.4397   |   0.1007   |   0.0590    |   0.2281   |   0.0723   |   0.0608   |   0.3489   |
> | LightGCN + InfoNCE |   0.1030   |   0.0976    |   0.5039   |   0.1094   |   0.0645    |   0.2454   |   0.0908   |   0.0785   |   0.4021   |
> |      XSimGCL       |   0.1057   |   0.0984    |   0.5128   |   0.1138   |   0.0662    |   0.2550   |   0.0879   |   0.0745   |   0.3918   |
> |    XSimGCL$ _{t}$     |   0.1015   |   0.0951    |   0.5016   |   0.1199   |   0.0679    |   0.2674   |   0.0900   |   0.0736   |   0.4036   |
> |   AlphaRec + BPR   |   0.0841   |   0.0746    |   0.4386   |   0.1081   |   0.0626    |   0.2447   |   0.0632   |   0.0516   |   0.3150   |
> | AlphaRec + InfoNCE | **0.1221** | **0.1144**  | **0.5587** | **0.1519** | **0.0894**  | **0.3207** | **0.0991** | **0.0828** | **0.4185** |
>
> The results reveal two key findings:
>
> 1. Language representations are powerful and have the potential to replace learnable IDs in recommender systems.
> This is evidenced by AlphaRec’s consistently strong results compared to both CL-based and LLM-enhanced CF methods, and its demonstrated abilities in zero-shot and intention-aware tasks in our paper.
>
> 2. However, without appropriate loss functions and model architecture, this potential cannot be fully realized:
> - Loss function: As you noted, AlphaRec+BPR performs similarly to LightGCN+BPR.
> This demonstrates that simply replacing learnable IDs with language representations in traditional models does not guarantee improvements.
> Notably, InfoNCE loss outperforms pair-wise BPR loss by better handling multiple negatives, which is crucial for learning collaborative signals in recommendation.
> This enables a more robust and powerful mapping from language space to behavior space, unlocking the potential of language representations.
> - Model architecture: Using language representations in XSimGCL without altering its architecture led to inconsistent performance improvement.
> We attribute this to the data augmentation technique in XSimGCL introducing noise, which may harm language representation quality.
>
> These findings suggest that unleashing the potential of language representations requires careful and **appropriate model design**, including both contrastive loss (e.g., InfoNCE) and a tailored architecture.
> Simply replacing input features of existing advanced CF models and observing a performance drop risks **underestimating the capabilities of language representations**.

---

> ### Author Response · Authors · 2024-11-23
> **Response to reviewer m7FA - Part (2/3)**
>
> ---
>
> > **Comment 2: The implementation of KAR and RLMRec should align with AlphaRec.**
>
> Thanks for your concern about the implementation of KAR and RLMRec.
> According to your suggestions, we have **revised the corresponding description of KAR and RLMRec and added one more ablation experiment about RLMRec in Appendix C.2.**
>
> In our paper, the implementation of KAR and RLMRec has been closely aligned with AlphaRec.
> For these methods, the CL is introduced.
> Moreover, we all use the same language representations as in AlphaRec (i.e., item representations from titles and user representations from the average of items).
>
> We have also tried another version of RLMRec, which is more similar to their original implementation (i.e., item and user profiles are generated by LLMs).
> For this version, we follow the same prompting approach in the original RLMRec paper to create user and item profiles.
>
> As reported in Table 1 below, the performance of this version of RLMRec (LLM-generated profile) is lower than the implementation in our submission.
> Moreover, the performance of incorporating LLM-generated profiles is not always stable, which may even lead to performance degradation in some cases (e.g., Movies & TV).
> We attribute the performance gap to the possible noise and hallucination introduced by these generated profiles.
> If the language representation is sufficiently powerful, as we expected, the item titles should already capture the necessary information about profiles, making additional LLM-generated profiles redundant or detrimental.
> We believe these results can make our conclusion more convincing.
>
> **Table 1 Performance comparison of different versions of RLMRec**
>
> |                                |        | Movies & TV |        |        | Video Games |        |        | **Books** |        |
> | :----------------------------: | :----: | :---------: | :----: | :----: | :---------: | :----: | :----: | :-------: | :----: |
> |                                | Recall |    NDCG     |   HR   | Recall |    NDCG     |   HR   | Recall |   NDCG    |   HR   |
> |            XSimGCL             | 0.1057 |   0.0984    | 0.5128 | 0.1138 |   0.0662    | 0.2550 | 0.0879 |  0.0745   | 0.3918 |
> | RLMRec (LLM-generated profile) | 0.1046 |   0.0942    | 0.5063 | 0.1218 |   0.0696    | 0.2692 | 0.0905 |  0.0741   | 0.4049 |
> |      RLMRec (Our Paper)       | 0.1119 |   0.1013    | 0.5301 | 0.1384 |   0.0809    | 0.2997 | 0.0928 |  0.0774   | 0.4092 |
>
> ---
>
> > **Comment 3: Evaluation on private and newly published datasets can help alleviateng the information leakage problem.**
>
> Thanks for your concern about the information leakage problem.
> Your suggestions about adopting private or new datasets are insightful.
> We have **revised our paper in Appendix C.6 to add one new dataset Amazon Electronic 2023 [1].**
>
> Collecting private datasets is difficult for academic research.
> Therefore, we consider the latest version of Amazon 2023 [1] as the newly published dataset.
> This dataset was published in March 2024, which is the latest public dataset that we have access to.
> To further alleviate the information leakage problem, we only consider the interactions after 2022.
>
> We conduct the same experiment as in Section 4 on this new dataset, and present the performance in Table 2 below.
> As shown in this table, in this new dataset, AlphaRec still consistently outperforms baselines.
> We attribute this result to the fact that language models understand the user preference behind item title descriptions rather than naively remembering the training data.
> Therefore, AlphaRec still works fine on the latest dataset.
>
> **Table 2 Performance of AlphaRec on Amazon Electronics 2023**
>
> |          |            | Amazon Electronics 2023 |            |
> | :------: | :--------: | :---------------------: | :--------: |
> |          |   Recall   |          NDCG           |     HR     |
> |    MF    |   0.0130   |         0.0089          |   0.0136   |
> | MultVAE  |   0.0227   |         0.0158          |   0.0237   |
> | LightGCN |   0.0237   |         0.0161          |   0.0248   |
> |   SGL    |   0.0519   |         0.0250          |   0.0551   |
> | BC Loss  |   0.0548   |         0.0265          |   0.0585   |
> | XSimGCL  |   0.0534   |         0.0261          |   0.0569   |
> |   KAR    |   0.0611   |         0.0283          |   0.0661   |
> |  RLMRec  |   0.0633   |         0.0288          |   0.0674   |
> | AlphaRec | **0.0687** |       **0.0323**        | **0.0732** |

---

> ### Author Response · Authors · 2024-11-23
> **Response to reviewer m7FA - Part (3/3)**
>
> ---
>
>
> > **Comment 4: To support the claim of collaborative signals encoded in LMs, the performance of using original language representations without any transformations should be considered.**
>
> Thanks for your valuable suggestions. **We have revised our paper in Section 4.2 accordingly.**
>
> We have added experiments using original language representations for recommendation.
> As shown in Table 3, directly using the language representations without training (i.e., EMB-KNN) consistently outperforms the heuristic method (i.e., Pop).
> Moreover, in most cases, EMB-KNN is comparable with MF and even surpasses it on Video Games.
> These findings are consistent with previous works [2].
> Such experimental results further help us understand the possibility of collaborative signals encoded within language representations.
>
> **Table 3 Performance of directly using language representations for recommendation**
>
> |          |        | Movies & TV |        |        | Video Games |        |        | **Books** |        |
> | :------: | :----: | :---------: | :----: | :----: | :---------: | :----: | :----: | :-------: | :----: |
> |          | Recall |    NDCG     |   HR   | Recall |    NDCG     |   HR   | Recall |   NDCG    |   HR   |
> |   Pop    | 0.0067 |   0.0058    | 0.0682 | 0.0084 |   0.0041    | 0.0248 | 0.0042 |  0.0018   | 0.0345 |
> |    MF    | 0.0568 |   0.0519    | 0.3377 | 0.0323 |   0.0195    | 0.0864 | 0.0437 |  0.0391   | 0.2476 |
> | LightGCN | 0.0849 |   0.0747    | 0.4397 | 0.1007 |   0.0590    | 0.2281 | 0.0722 |  0.0597   | 0.3418 |
> | EMB-KNN  | 0.0548 |   0.0380    | 0.2916 | 0.0879 |   0.0389    | 0.1970 | 0.0434 |  0.0248   | 0.1851 |
>
> [1] Hou, Yupeng, et al. "Bridging language and items for retrieval and recommendation." *arXiv preprint arXiv:2403.03952* (2024).
>
> [2] Ding, Hao, et al. "Zero-shot recommender systems." *arXiv preprint arXiv:2105.08318* (2021).

---

> ### Author Response · Authors · 2024-11-25
> **Looking forward to your reply**
>
> Dear Reviewer m7fA.
>
> Thank you again for your positive feedback and insightful questions. Following your suggestions, we have **added the necessary discussion and experiments about the comparison fairness**. Moreover, to address the possible information leakage problem you have mentioned, we **conduct experiments on the latest dataset Amazon Electronics 2023**. We also **present the performance of directly adopting language representations for recommendation** to better showcase their power. About **revisions have been added to the latest version** of our paper.
>
> We look forward to further discussion with you.
>
> Best regards,
>
> Authors.

---

> > ### Comment · Reviewer_m7fA · 2024-11-28
> >
> > Thanks for the detailed responses from authors.

---

> > > ### Author Response · Authors · 2024-11-28
> > > **Thanks!**
> > >
> > > We sincerely thank you for your time and valuable suggestions! Your comments have significantly helped us refine our paper.

---

### Official Review · Reviewer_uzkZ · 2024-10-27

**Soundness:** 3
**Presentation:** 4
**Contribution:** 3
**Rating:** 8
**Confidence:** 4

**Summary:**

This paper studies the potential of leveraging advanced language models (LMs) for recommendation tasks, exploring whether language representations can encode user preferences and collaborative signals. Traditionally, CF-based recommendation systems use ID-based embeddings to model user-item interactions. The authors propose a method that directly maps language representations of item metadata (like titles) into a behavior space to serve as effective item representations. They introduce a linear mapping approach, followed by the AlphaRec model, which combines language embeddings with collaborative filtering elements such as graph convolution and contrastive loss.

Generally, this is a very good paper provides new insights in the community: homomorphism exists between LM- based language space and user behavior space for recommendation.

**Strengths:**

1. The topic on studying the relationship between the language space and behavior space is important to the community. The authors provide new insights to show the homomorphism between the two spaces.

2. The paper’s core contribution, mapping language representations into a behavior space, is thoroughly tested and demonstrates strong empirical performance. This novel approach could reduce reliance on large datasets of historical behavior data, making the method highly applicable.

3. Followed by the observation through linear mapping, the authors also propose a stronger model, AlphaRec, through non-linear mapping and GNN. Its ability to perform zero-shot recommendations and its efficient training process are well-demonstrated in multiple datasets, addressing common challenges in recommendation systems where user or item IDs change frequently.

4. The paper’s evaluation covers a range of experimental setups, comparing multiple LMs, testing robustness to input noise, and conducting ablation studies. The discussion on the potential of language-based user representation (Section 5) is promising.

**Weaknesses:**

1. While the paper presents a compelling case for using language representations in recommendation, a discussion on how AlphaRec compares to existing fine-tuned LLM approaches would strengthen it further. For example, it would be helpful to understand whether the authors see AlphaRec as complementary or competitive with fine-tuned LLM-based approaches (e.g., [1]), and if they anticipate similar benefits, limitations, or unique strengths compared to those models.

[1] Tennenholtz et al. Demystifying Embedding Spaces using Large Language Models. ICLR 2024.


2. The authors assume that a single linear mapping can capture collaborative signals for all users, which may overlook the diversity in individual user preferences and linguistic nuances. Different users might interpret language in varied ways, and a one-size-fits-all mapping could limit the model’s ability to fully capture such personalization and diversity. How might the authors address this challenge within their framework? Exploring the possibility of user-specific mappings or personalization approaches could be a valuable direction for future work, potentially enhancing the model’s adaptability across diverse user groups.

**Questions:**

The discussion on the potential of intent-aware ability is interesting. However, as indicated in Appendix D.3.1, a fixed user intention query is generated for each item in the dataset. I would argue that this is not optimal, as different users may have different intents even when consuming the same item. Could the authors consider extending their approach to allow for dynamic query generation or user-specific intent modeling, potentially improving recommendation relevance?

---

> ### Author Response · Authors · 2024-11-23
> **Response to reviewer uzkZ - Part (1/2)**
>
> Thanks for your positive feedback and acknowledgment of the importance of the research topic!
> Your suggestions have greatly helped us strengthen our submission.
>
> ---
>
> > **Comment 1: Discussion with existing fine-tuned LLM-based approaches can strengthen the claims in this paper.**
>
> Thanks for bringing this high-quality paper and for your valuable comments!
> We fully agree that fine-tuned LLM-based approaches are highly inspiring and deserve further discussion.
> Following your suggestion, we **added ELM [1] as a reference in the introduction section** and added a **detailed discussion in Appendix B.3** of the revised paper.
>
> We would like to categorize the research line (i.e., fine-tuned LLM approaches), including ELM [1], as efforts to understand domain-specific representations by projecting them into an LLM and fine-tuning the LLM to generate explanations.
> This line of research also includes works we mentioned in our original paper in Line 77-79, such as RecExplainer [2] and RecInterpreter [3], which use LLMs for interpretability, and LLaRA and CoLLM, which utilize LLMs for direct recommendations.
> In these fine-tuned LLM approaches, the behavior representations are projected as a new modality into the token space of LLMs.
> The fact that LLMs can interpret such behavior representations indicates that the **user behavior space learnt from domain-specific dataset can be aligned to language space at the token level**.
>
> AlphaRec and these approaches share a complementary goal: exploring the connection between language modeling and user behavior modeling from a representation space perspective.
> However, they **differ in strengths and limitations**:
>
> Fine-tuned LLM-based approaches:
> - Strengths:
> 1. Inject domain-specific representations into LLMs, allowing for interpreting the meaning of representations from the behavior space in a generative way.
> 2. Offer potential for user-friendly, explainable recommender systems, which is the future from my personal perspective.
> - Limitations:
> 1. Domain-specific representations inherently carry the limitations of their domain-specific nature, often lacking the generalization ability required to effectively address zero-shot challenges in recommendations.
> 1. High inference time, which can hinder online implementation.
>
> AlphaRec:
>
> - Strengths:
> 1. Directly leverages language representations, allowing for easy adaptation to real-world recommendation platforms as a plug-and-play, low-cost solution.
> 2. Exhibits potential to capture short-term user intentions efficiently.
> - Limitations:
> Lacks the intuitive interpretability for human understanding, as it does not generate natural language explanations.
>
> We believe both research directions are highly significant and will continue to inspire each other.
> If you have additional insights, we would be delighted to discuss them further with you.
> Thank you again for your thoughtful question!

---

> ### Author Response · Authors · 2024-11-23
> **Response to reviewer uzkZ - Part (2/2)**
>
> ---
>
> > **Comment 2: Personalized user-specific mapping is worth exploring.**
>
> Your suggestion to incorporate personalization into AlphaRec is insightful!
> It is true that while AlphaRec ensures the generalization across all users through a shared MLP, it overlooks the personalization of individual users, which is a current limitation and worth exploring.
> In response, **we have revised the limitation (i.e., Section 6) to include a discussion on this limitation on personalization.**
>
> Moreover, inspired by your suggestion, we explore a minimal personalization approach by introducing a learnable parameter $p_{u}$ for each user (i.e., replacing $z_{u}$ with $p_{u}z_{u}$) to examine the effectiveness of personalization.
> We denote this adjusted model as AlphaRec-p.
> As shown in Table 1 below, this modification results in slight performance improvements on two datasets, demonstrating that even minimal personalization can bring benefits.
>
> We also believe that more diverse and advanced personalization approaches, such as Mixture of Experts (MoE) [4], are worth exploring in the future.
>
> **Table 1 Performance of personalized AlphaRec**
>
> |            |            | Movies & TV |            |            | Video Games |            |            | **Books**  |            |
> | :--------: | :--------: | :---------: | :--------: | :--------: | :---------: | :--------: | :--------: | :--------: | :--------: |
> |            |   Recall   |    NDCG     |     HR     |   Recall   |    NDCG     |     HR     |   Recall   |    NDCG    |     HR     |
> |  AlphaRec  | **0.1221** |   0.1144    | **0.5587** |   0.1519   |   0.0894    |   0.3207   |   0.0991   |   0.0828   |   0.4185   |
> | AlphaRec-p |   0.1219   | **0.1156**  |   0.5564   | **0.1525** | **0.0897**  | **0.3215** | **0.1001** | **0.0834** | **0.4202** |
>
>
> ---
>
> > **Comment 3: Dynamic query generation for user-specific intent modeling is worth exploring.**
>
> Thanks for your suggestion!
> We **conducted an addtional experiment** and **revised our paper accordingly.**
>
> The goal of this section is to explore the potential of perceiving user intention.
> Therefore, we simplify the problem by only considering a fixed intention.
> However, it is true that user-specific intention is more practical in the real-world scenarios.
> Following your valuable suggestions, we consider generating dynamic and personalized intentions by adding the user's historical interactions in the prompt (i.e., "The user has interacted with the following items: <item_list>").
> This naive personalized user intention method is denoted as AlphaRec (dynamic intention), while the original fixed intention is denoted as AlphaRec (static intention).
> As shown in Table 2 below, by considering the dynamic intention, the recommendation results are refined more precisely.
> We believe that incorporating personalized intentions is a very important and practical research problem deserving further exploration.
>
> **Table 2 Intention capture results when adopting dynamic user intention**
>
> |                              | MovieLens-1M |        |
> | :--------------------------: | :----------: | :----: |
> |                              |     HR@5     | NDCG@5 |
> | AlphaRec (static intention)  |    0.4704    | 0.3738 |
> | AlphaRec (dynamic intention) |    0.5118    | 0.4099 |
>
> [1] Tennenholtz et al. Demystifying Embedding Spaces using Large Language Models. ICLR 2024.
>
> [2] Lei, Yuxuan, et al. "RecExplainer: Aligning Large Language Models for Explaining Recommendation Models." KDD. 2024.
>
> [3] Yang, Zhengyi, et al. "Large language model can interpret latent space of sequential recommender." *arXiv preprint arXiv:2310.20487* (2023).
>
> [4] Hou, Yupeng, et al. "Towards universal sequence representation learning for recommender systems." *Proceedings of the 28th ACM SIGKDD Conference on Knowledge Discovery and Data Mining*. 2022.

---

> ### Author Response · Authors · 2024-11-25
> **Looking forward to your reply**
>
> Dear Reviewer uzkZ.
>
> Thank you again for your positive feedback and valuable suggestions. According to your suggestions, we have added the necessary **discussion with fine-tuned LLM approaches**. Moreover, we have also explored **the personalization of AlphaRec and intention query generation**. We have also **revised our paper**, adding discussions about these topics.
>
> We look forward to further discussion with you.
>
> Best regards,
>
> Authors.

---

> > ### Comment · Reviewer_uzkZ · 2024-11-27
> > **Thanks for your rebuttal**
> >
> > I appreciate the authors for the response during rebuttal. I think the responses solved most of my concerns, so I increased my score. I would suggest the authors properly add these responses into the final manuscript if possible.

---

> > > ### Author Response · Authors · 2024-11-28
> > > **Thanks for your review!**
> > >
> > > Thanks for your positive comments and feedback! Your suggestions have significantly helped us refine our work. We will add the necessary revisions in our latest version!

---

### Official Review · Reviewer_QRqF · 2024-11-04

**Soundness:** 3
**Presentation:** 3
**Contribution:** 3
**Rating:** 5
**Confidence:** 5

**Summary:**

The paper challenges the idea that Language Models (LMs) and traditional recommenders exist in separate representation spaces. It proposes deriving a recommendation space directly from language representations. By mapping item representations from advanced LM outputs, it shows improved recommendation performance, hinting at a homomorphism between language and item representation spaces. The paper introduces collaborative filtering models based solely on language representations, excluding ID-based embeddings. Components like multilayer perceptrons, graph convolution, and contrastive learning are integrated into the model.

**Strengths:**

1. The authors propose using linear mapping of LLMs' representations to directly model user preferences, which is both efficient and effective.

2. The authors conducted comprehensive experiments discussing the performance of AlphaRec in full-shot, zero-shot, and other scenarios, comparing it with baseline methods from various research lines.

**Weaknesses:**

1. From Table 1, it can be observed that the performance of AlphaRec is heavily dependent on the performance of the chosen language model. When selecting smaller language models (such as BERT, RoBERTa, etc.), the performance is less than ideal. Additionally, the costs associated with state-of-the-art methods (i.e., SFR-Embedding-Mistral, text-embeddings-3-large, etc.), including inference costs or API call expenses, cannot be overlooked, which weakens the applicability of this method.

2. The authors omitted a method that performs well in zero-shot scenarios, VQ-Rec [1], when exploring zero-shot capabilities.

3. In the ablation study section, it can be observed that AlphaRec without Contrastive Learning (CL) performs significantly worse than Linear Mapping, indicating that simply adding MLP and GCN on top of Linear Mapping does not lead to effective performance improvements.

4. If LLMs' representations and MLP are applied to state-of-the-art ID-based methods (e.g., XSimGCL) to replace the original ID embeddings, would it achieve results comparable to those of AlphaRec?

[1] Hou, Yupeng, et al. "Learning vector-quantized item representation for transferable sequential recommenders." Proceedings of the ACM Web Conference 2023. 2023.

**Questions:**

Please refer to the issues mentioned in the Weaknesses section.

---

> ### Author Response · Authors · 2024-11-23
> **Response to reviewer QRqF - Part (1/3)**
>
> Thanks for your acknowledgment of the linear mapping method in our paper and the experimental comprehensiveness.
> We also highly appreciate your concerns and valuable suggestions about our paper.
> Your suggestions have greatly helped us refine our paper.
> We respond to your questions as follows:
>
> ---
>
> > **Comment 1: The performance of AlphaRec heavily relies on the language model. The inference cost and API cost can not be overlooked.**
>
> Thanks for your valuable concerns about the performance and cost of AlphaRec with different language representations.
> We carefully **revised our paper** and tried to address your concerns as follows:
>
> Firstly, we want to clarify that the non-linearity and neighborhood aggregation introduced in AlphaRec narrow the performance gap between different language representations.
> According to the suggestion of yours and reviewer **V7NB**, we have **conducted an ablation study** to analyze the effect of varying the input language representations for AlphaRec.
> As shown in Table 1 below, the performance gap of AlphaRec between different language representations narrows, compared with the linear mapping setting in our original paper.
>
> Secondly, we would like to point out that the inference cost of LLMs and API cost are relatively affordable, since we only need to extract the language representations of item titles for one time.
> Extracting the representations with a 7B model on one single V100 gpu for 10000 items only takes around 6 minutes, and using the API to generate representations for 10000 items costs less than 0.2 US dollars. We have revised the paper to add Table 16 about the cost of extracting features.
> We think these costs are reasonable and affordable.
> After extracting these representations, we can store them and train a language-representation-based recommender.
> In this stage, the training cost is similar to traditional ID-based recommenders, as illustrated in Figure (3b) of the paper.
> The inference cost is relatively acceptable since the language representations are frozen.
>
> **Table 1 Ablation study of varying language representations for AlphaRec**
>
> |                         |            | Movies & TV |            |            | Video Games |            |            | **Books**  |            |
> | :---------------------: | :--------: | :---------: | :--------: | :--------: | :---------: | :--------: | :--------: | :--------: | :--------: |
> |        AlphaRec         |   Recall   |    NDCG     |     HR     |   Recall   |    NDCG     |     HR     |   Recall   |    NDCG    |     HR     |
> |          BERT           |   0.0994   |   0.0923    |   0.4873   |   0.0960   |   0.0550    |   0.2179   |   0.0719   |   0.0607   |   0.3434   |
> |         RoBERTa         |   0.0967   |   0.0895    |   0.4793   |   0.0947   |   0.0545    |   0.2167   |   0.0710   |   0.0596   |   0.3386   |
> |        Llama2-7B        |   0.1160   |   0.1092    |   0.5388   |   0.1395   |   0.0817    |   0.3003   |   0.0940   |   0.0793   |   0.4081   |
> |       Mistral-7B        |   0.1161   |   0.1097    |   0.5421   |   0.1413   |   0.0828    |   0.3020   |   0.0945   |   0.0799   |   0.4090   |
> |  text-embedding-ada-v2  |   0.1152   |   0.1083    |   0.5382   |   0.1437   |   0.0844    |   0.3062   |   0.0933   |   0.0784   |   0.4061   |
> | Text-embeddings-3-large |   0.1221   | **0.1144**  | **0.5587** |   0.1519   | **0.0894**  |   0.3207   | **0.0991** | **0.0828** | **0.4185** |
> |  SFR-Embedding-Mistral  | **0.1225** |   0.1139    |   0.5571   | **0.1521** |   0.0887    | **0.3209** |   0.0982   |   0.0820   |   0.4161   |

---

> ### Author Response · Authors · 2024-11-23
> **Response to reviewer QRqF - Part (2/3)**
>
> ---
> > **Comment 2: VQ-Rec is omitted in the zero-shot experiment.**
>
> Thanks for your insightful suggestion about the baseline of VQ-Rec.
> We have **added the implementation of VQ-Rec in the zero-shot experiment part** and **revised our paper in Section 5.2**.
> We report the updated zero-shot performance in the following Table 2.
> AlphaRec presents relatively high zero-shot performance compared with baselines across various datasets, and VQ-Rec presents improvements compared with ZESRec and UnisRec.
> Thanks for your suggestion.
>
> **Table 2. VQ-Rec performance in zero-shot settings**
>
> |          |            | Industrial & Scientific |            |            | MovieLens-1M |            |            | **Book Crossing** |            |
> | :------: | :--------: | :---------------------: | :--------: | :--------: | :----------: | :--------: | :--------: | :---------------: | :--------: |
> |          |   Recall   |          NDCG           |     HR     |   Recall   |     NDCG     |     HR     |   Recall   |       NDCG        |     HR     |
> |  ZESRec  |   0.0326   |         0.0272          |   0.0628   |   0.0274   |    0.0787    |   0.5786   |   0.0155   |      0.0143       |   0.1347   |
> | UniSRec  |   0.0453   |         0.0350          |   0.0863   |   0.0578   |    0.1412    |   0.7135   |   0.0396   |      0.0332       |   0.2454   |
> |  VQ-Rec  |   0.0645   |         0.0410          |   0.0963   |   0.0804   |    0.1921    |   0.8167   |   0.0485   |      0.0492       |   0.2825   |
> | AlphaRec | **0.0913** |       **0.0573**        | **0.1277** | **0.1486** |  **0.3215**  | **0.9296** | **0.0660** |    **0.0545**     | **0.3381** |
>
> ---
>
> > **Comment 3: Simply adding MLP and GCN on the top of linear mapping does not lead to effective performance improvements.**
>
> Thanks for your careful reading and valuable concerns.
> We might not fully understand your question, and we are happy to discuss it with you if our response does not address your concerns.
>
> In the linear mapping experiments, the CL has been adopted.
> CL has been proven to be one of the most important components for recommendation [1, 2, 3].
> Therefore, the results in the ablation study that linear mapping with CL outperforms AlphaRec with GCN + MLP +BPR are reasonable and consistent with these previous findings [1, 2, 3].
> Moreover, the difference between AlphaRec and linear mapping lies in the MLP and GCN, and the performance improvements exactly indicate the significant roles of these components.
> This phenomenon further highlights the importance of **appropriate model design** for stimulating the potential of language representations, as we have concluded in Section 4.2.

---

> ### Author Response · Authors · 2024-11-23
> **Response to reviewer QRqF - Part (3/3)**
>
> ---
>
> > **Comment 4: What is the performance of replacing the ID embeddings in XSimGCL with language representations?**
>
> Thanks for your question.
> Following your suggestion, we have **conducted additional experiments** replacing the ID embeddings in XSimGCL with language representations.
> We have also **revised our paper in Section 4.2** according to your suggestion.
>
> We denote this model as XSimGCL$ _ {t}$.
> As shown in Table 2, we find that the performance of XSimGCL$ _ {t}$ improves on some datasets but declines in others.
> We attribute this to a possible reason that the data augmentation technique by adding noise in XSimGCL may harm the language representations.
> This unstable performance phenomenon also aligns with existing studies [4, 5], which show that even the most advanced ID-based recommenders cannot achieve consistent performance gains through a naive input feature replacement with language representations alone.
> These findings further emphasize the **importance of careful and appropriate model design** to stimulate the potential of language representations for recommendation.
>
> **Table 2 Performance comparison when replacing ID embeddings with language representations in XSimGCL**
>
> |             |            | Movies & TV |            |            | Video Games |            |            | **Books**  |            |
> | :---------: | :--------: | :---------: | :--------: | :--------: | :---------: | :--------: | :--------: | :--------: | :--------: |
> |             |   Recall   |    NDCG     |     HR     |   Recall   |    NDCG     |     HR     |   Recall   |    NDCG    |     HR     |
> |   XSimGCL   |   0.1057   |   0.0984    |   0.5128   |   0.1138   |   0.0662    |   0.2550   |   0.0879   |   0.0745   |   0.3918   |
> | XSimGCL$ _ {t}$  |   0.1015   |   0.0951    |   0.5016   |   0.1199   |   0.0679    |   0.2674   |   0.0900   |   0.0736   |   0.4036   |
> |  AlphaRec   | **0.1221** | **0.1144**  | **0.5587** | **0.1519** | **0.0894**  | **0.3207** | **0.0991** | **0.0828** | **0.4185** |
>
> [1] Yu, Junliang, et al. "Are graph augmentations necessary? simple graph contrastive learning for recommendation." *Proceedings of the 45th international ACM SIGIR conference on research and development in information retrieval*. 2022.
>
> [2] Yu, Junliang, et al. "XSimGCL: Towards extremely simple graph contrastive learning for recommendation." *IEEE Transactions on Knowledge and Data Engineering* 36.2 (2023): 913-926.
>
> [3] Wu, Jiancan, et al. "On the effectiveness of sampled softmax loss for item recommendation." *ACM Transactions on Information Systems* 42.4 (2024): 1-26.
>
> [4] Yuan, Zheng, et al. "Where to go next for recommender systems? id-vs. modality-based recommender models revisited." *Proceedings of the 46th International ACM SIGIR Conference on Research and Development in Information Retrieval*. 2023.
>
> [5] Li, Ruyu, et al. "Exploring the upper limits of text-based collaborative filtering using large language models: Discoveries and insights." *arXiv preprint arXiv:2305.11700* (2023).

---

> ### Author Response · Authors · 2024-11-25
> **Looking forward to your reply**
>
> Dear Reviewer QRqF,
>
> Thank you again for your insightful suggestions. According to your suggestions, we have added necessary **discussion about the cost of extracting features**. Moreover, we have **added the VQ-Rec** under the zero-shot recommendation scenario. We also add a **discussion about the role of MLP and GCN**, and present the results of **replacing ID embeddings with language representations in XSimGCL**. All the **revisions have been incorporated** in our latest version of paper.
>
> We look forward to further discussion with you.
>
> Best regards,
>
> Authors

---

### Official Review · Reviewer_V7NB · 2024-11-05

**Soundness:** 4
**Presentation:** 4
**Contribution:** 4
**Rating:** 8
**Confidence:** 2

**Summary:**

This paper explores the relationship between language space and behavior space in the context of recommendation systems, investigating the potential of using language representations for this purpose. The proposed method, AlphaRec, demonstrates excellent performance in zero-shot recommendation scenarios.

**Strengths:**

1. This paper is well-written and easy to follow.
2. Leveraging textual information for recommendations and exploring zero-shot performance is both novel and interesting.
3. The experiments are comprehensive and effectively demonstrate the proposed method's effectiveness.

**Weaknesses:**

1. It would be beneficial to conduct an additional ablation study to evaluate different types of text embedding models on AlphaRec.

**Questions:**

See Weaknesses.

---

> ### Author Response · Authors · 2024-11-23
> **Response to reviewer V7NB**
>
> We sincerely appreciate your positive feedback and valuable suggestions!
> Your suggestion about the ablation study on the adopted language representations in AlphaRec is important.
> We have **conducted suggested experiments and revised the paper** accordingly in Appendix C.4.
>
> ---
>
> > **Comment 1: Ablation study of different text embedding models for AlphaRec can be considered.**
>
> Following your suggestions, we have added an ablation study varying language representations for AlphaRec and reported the results in Table 1 below.
> The results reveal a clear trend: advanced language representations, such as SFR-Embedding-Mistral and Text-embeddings-3-large, consistently outperform earlier BERT-style models across all datasets. This finding aligns with the observations from the linear mapping experiments, further reinforcing the critical role of high-quality language representations in recommendation tasks.
> Additionally, AlphaRec's architecture design, with its incorporation of non-linearity and neighborhood aggregation, reduces reliance on specific language models.
> This design narrows the performance gap between different language representations, as compared to the linear mapping setting in our original paper.
>
>
> **Table 1 Ablation study of varying language representations for AlphaRec**
>
> |                         |            | Movies & TV |            |            | Video Games |            |            | **Books**  |            |
> | :---------------------: | :--------: | :---------: | :--------: | :--------: | :---------: | :--------: | :--------: | :--------: | :--------: |
> |        AlphaRec         |   Recall   |    NDCG     |     HR     |   Recall   |    NDCG     |     HR     |   Recall   |    NDCG    |     HR     |
> |          BERT           |   0.0994   |   0.0923    |   0.4873   |   0.0960   |   0.0550    |   0.2179   |   0.0719   |   0.0607   |   0.3434   |
> |         RoBERTa         |   0.0967   |   0.0895    |   0.4793   |   0.0947   |   0.0545    |   0.2167   |   0.0710   |   0.0596   |   0.3386   |
> |        Llama2-7B        |   0.1160   |   0.1092    |   0.5388   |   0.1395   |   0.0817    |   0.3003   |   0.0940   |   0.0793   |   0.4081   |
> |       Mistral-7B        |   0.1161   |   0.1097    |   0.5421   |   0.1413   |   0.0828    |   0.3020   |   0.0945   |   0.0799   |   0.4090   |
> |  text-embedding-ada-v2  |   0.1152   |   0.1083    |   0.5382   |   0.1437   |   0.0844    |   0.3062   |   0.0933   |   0.0784   |   0.4061   |
> | Text-embeddings-3-large |   0.1221   | **0.1144**  | **0.5587** |   0.1519   | **0.0894**  |   0.3207   | **0.0991** | **0.0828** | **0.4185** |
> |  SFR-Embedding-Mistral  | **0.1225** |   0.1139    |   0.5571   | **0.1521** |   0.0887    | **0.3209** |   0.0982   |   0.0820   |   0.4161   |
>
> We sincerely appreciate your suggestion, which has helped us deepen our analysis and strengthen the claims in our paper. Thank you!

---

> ### Author Response · Authors · 2024-11-25
> **Looking forward to your reply**
>
> Dear Reviewer V7NB,
>
> Thank you again for your positive comments and valuable suggestions. We have **revised our paper** according to your suggestions, adding one **ablation study on the language representations of AlphaRec.**
>
> We look forward to further discussion with you.
>
> Best regards,
> Authors

---

### Meta-Review · Area_Chair_kQmi · 2024-12-19

**Metareview:**

This paper explores the relationship between language space derived from LMs and user-behavior space in RecSys, investigating whether language representations can encode user behaviors, especially collaborative signals. The proposed method, AlphaRec, integrates language representations with collaborative filtering, showcasing strong performance in zero-shot and intention-aware recommendation scenarios.
Strengths:
- The paper tackles a meaningful and underexplored problem, providing insights into the homomorphism between language and behavior spaces in recommendation.
- The proposed method AlphaRec addresses common challenges in recommendation systems, such as frequent changes in user or item IDs, and highlights the potential of language-based models for scalable, zero-shot recommendation tasks.
- The performance of AlphaRec is thoroughly evaluated across diverse datasets and scenarios, including full-shot, zero-shot, and intention-aware recommendations. Comprehensive experiments include comparisons with various baseline methods and ablation studies, providing robust evidence of its effectiveness.

Overall, the paper presents the understanding and application of language models in recommendation systems well.

**Additional Comments On Reviewer Discussion:**

The authors actively discussed with the reviewers by conducting additional experiments and revising the paper:
• Added experiments analyzing the performance and cost of AlphaRec with different language representations (Reviewer V7NB & QRqF & m7fA)
• Included a new baseline, VQ-Rec, in zero-shot recommendation experiments, revising Section 5.2 to reflect these results (Reviewer QRqF)
• Added discussions on the roles of MLP and GCN in the proposed method (Reviewer QRqF)
• Presented results of replacing ID embeddings with language representations in XSimGCL (Reviewer QRqF)
• Added discussions on the cost of feature extraction and its implications for AlphaRec's performance and efficiency (Reviewer QRqF)
• Incorporated discussions on fine-tuned LLM approaches
• Explored the personalization capabilities of AlphaRec and the generation of intention queries (Reviewer uzkZ)

All these updates have been incorporated into the revised version of the paper, addressing reviewer feedback comprehensively.

---

### Decision · Program_Chairs · 2025-01-22

Accept (Oral)